# ODAM: Gradient-based Instance-Specific Visual Explanations for Object Detection

**Chenyang Zhao & Antoni B. Chan**
Department of Computer Science
City University of Hong Kong
`chenyzhao9-c@my.cityu.edu.hk, abchan@cityu.edu.hk`

## Abstract

We propose the gradient-weighted Object Detector Activation Maps (ODAM), a visualized explanation technique for interpreting the predictions of object detectors. Utilizing the gradients of detector targets flowing into the intermediate feature maps, ODAM produces heat maps that show the influence of regions on the detector's decision for each predicted attribute. Compared to previous works classification activation maps (CAM), ODAM generates instance-specific explanations rather than class-specific ones. We show that ODAM is applicable to both one-stage detectors and two-stage detectors with different types of detector backbones and heads, and produces higher-quality visual explanations than the state-of-the-art both effectively and efficiently. We next propose a training scheme, Odam-Train, to improve the explanation ability on object discrimination of the detector through encouraging consistency between explanations for detections on the same object, and distinct explanations for detections on different objects. Based on the heat maps produced by ODAM with Odam-Train, we propose Odam-NMS, which considers the information of the model's explanation for each prediction to distinguish the duplicate detected objects. We present a detailed analysis of the visualized explanations of detectors and carry out extensive experiments to validate the effectiveness of the proposed ODAM.

## 1 Introduction

Significant breakthroughs have been made in object detection and other computer vision tasks due to the development of deep neural networks (DNN) (Girshick et al., 2014b). However, the unintuitive and opaque process of DNNs makes them hard to interpret. As spatial convolution is a frequent component of state-of-the-art models for vision tasks, class-specific attention has emerged to interpret CNNs, which has been used to identify failure modes (Agrawal et al., 2016; Hoiem et al., 2012), debug models (Koh & Liang, 2017) and establish appropriate users' confidence about models (Selvaraju et al., 2017). These explanation approaches produce heat maps locating the regions in the input images that the model looked at, representing the influence of different pixels on the model's decision. Gradient visualization (Simonyan et al., 2013), Perturbation (Ribeiro et al., 2016), and Class Activation Map (CAM) (Zhou et al., 2016) are three widely adopted methods to generate the visual explanation map. However, these methods have primarily focused on image classification (Petsiuk et al., 2018; Fong & Vedaldi, 2017; Selvaraju et al., 2017; Chattopadhay et al., 2018; Wang et al., 2020b;a), or its variants, e.g., visual question answering (Park et al., 2018), video captioning (Ramanishka et al., 2017; Bargal et al., 2018), and video activity recognition (Bargal et al., 2018).

Generating explanation heat maps for object detectors is an under-explored area. The first work in this area is D-RISE (Petsiuk et al., 2021), which extends RISE (Petsiuk et al., 2018) for explaining image classifiers to object detectors. As a perturbation-based approach, D-RISE first randomly generates a large number of binary masks, resizes them to the image size, and then perturbs the original input to observe the change in the model's prediction. However, the large number of inference calculations makes the D-RISE computationally intensive, and the quality of the heat maps is influenced by the mask resolution (e.g., see Fig. 1b). Furthermore, D-RISE only generates an overall heat map for the predicted object, which is unable to show the influence of regions on the specific attributes of a prediction, e.g., class probability and regressed bounding box corner coordinates.

The popular CAM-based methods for image classification are not directly applicable to object detectors. CAM methods generate heat maps for classification via a linear combination of the weights and the activation maps, such as the popular Grad-CAM (Selvaraju et al., 2017) and its variants.

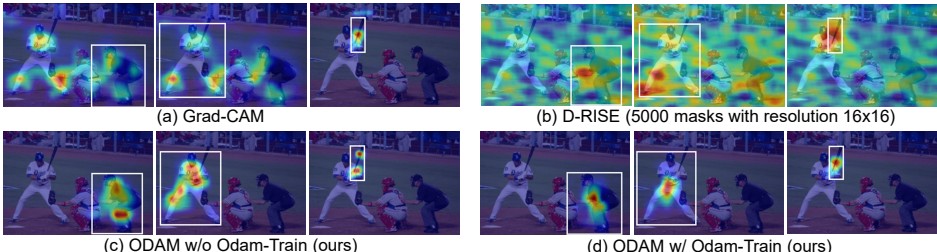

Figure 1: Comparison of heat maps from Grad-CAM (Selvaraju et al., 2017), D-RISE (Petsiuk et al., 2021) and our ODAM. The white box shows the corresponding detected object. (a) Grad-CAM highlights all objects of the same category (person) instead of the specific object instance. (b) D-RISE maps have noisy backgrounds and its effectiveness depends on the mask size; the 16x16 mask is better for smaller objects (baseball bat) than larger objects (person). (c) ODAM generates instance-specific heat maps with less noise and is robust to object size. (d) With Odam-Train, the heat map is better localized over the object and separated from other objects.

However, Grad-CAM provides class-specific explanations, and thus produces heat maps that highlight all objects of in a category instead of explaining a single detection (e.g., see Fig. 1a). For object detection, the explanations should be *instance-specific* rather than class-specific, so as to discriminate each individual object. Exploring the spatial importance of different objects can help interpret the models' decision and show the important area in the feature maps for each prediction.

Considering that direct application of existing CAM methods to object detectors is infeasible and the drawbacks of the current state-of-the-art D-RISE, we propose gradient-weighted *Object Detector Activation Maps* (ODAM). ODAM adopts a similar assumption as Grad-CAM that feature maps correlate with some concept for making the final outputs. Thus ODAM uses the gradients w.r.t. each pixel in the feature map to obtain the explanation heat map for each attribute of the object prediction. Compared with the perturbation-based D-RISE, ODAM is more efficient and generates less noisy heat maps (see Fig. 1c), while also explaining each attribute separately.

We also explore a unique explanation task for object detectors, *object discrimination*, which aims to explain *which* object was detected. This is different from the traditional explanation task of what features are important for class prediction (i.e., *object specfication*). We propose a training scheme, Odam-Train, to improve the explanation ability for object discrimination by introducing consistency and separation losses. The training encourages the model to produce consistent heat maps for the same object, and distinctive heat maps for different objects (see Fig. 1d). We further propose Odam-NMS, which uses the instance-level heat maps from ODAM to aid the non-maximum suppression (NMS) process of removing duplicate predictions of the same object.

The contributions of our paper are summarized as follows:

1. We propose ODAM, a gradient-based visual explanation approach to produce instance-specific heat maps for explaining prediction attributes of object detectors, which is more efficient and robust compared with the current state-of-the-art.
2. We demonstrate the generalizability of ODAM by exhibiting explanations on one- and two-stage, and transformer-based detectors with different types of backbones and detector heads.
3. We explore a unique explanation task for detector, object discrimination, for explaining which object was detected, and propose Odam-Train to obtain model with better object discrimination ability.
4. We propose Odam-NMS, which uses the instance-level heat maps generated by ODAM with Odam-Train to remove duplicate predictions during NMS, and its effectiveness verifies the object discrimination ability of ODAM with Odam-Train.

## 2 RELATED WORKS

**Object detection** Object detectors are generally composed of a backbone, neck and head. Based on the type of head, detectors can be mainly divided into one-stage and two-stage methods. Two-stage approaches perform two steps: generating region candidates (proposals) and then using RoI (Region of Interest) features for the subsequent object classification and location regression. The representative two-stage works are the R-CNN family, including R-CNN (Girshick et al., 2014a), Fast R-CNN (Girshick, 2015), Faster-RCNN (Ren et al., 2015), and Mask R-CNN (He et al., 2017). One-stage methods remove the RoI feature extraction and directly perform classification and regression on the entire feature map, and typical methods are YOLO (Redmon et al., 2016), RetinaNet (Lin et al., 2017b), and FCOS (Tian et al., 2019). Our ODAM can generate heat maps for both

one- and two-stage detectors with no limitation of the types of backbones and heads, as well as transformer-based detectors. We mainly adopt Faster R-CNN and FCOS in our experiments.

**Explanation by visualization** Since visualizing the importance of input features is a straightforward approach to interpret a model, many works visualize the internal representations of image classifier CNNs with heat maps. Gradient visualization methods (Simonyan et al., 2013) backpropagate the gradient of a target class score to the input image to highlight the "important" pixels, and other works (Springenberg et al., 2014; Zeiler & Fergus, 2014; Adebayo et al., 2018a) manipulate this gradient to improve the results qualitatively (see comparison in (Mahendran & Vedaldi, 2016)). The visualizations are fine-grained but not class-specific. Perturbation-based methods (Petsiuk et al., 2018; Ribeiro et al., 2016; Lundberg & Lee, 2017; Dabkowski & Gal, 2017; Chang et al., 2018; Fong & Vedaldi, 2017; Wagner et al., 2019; Lee et al., 2021) perturb the original input and observe the changes in output scores to determine the importance of regions. Most black-box methods are intuitive and highly generalizable, but computationally intensive. Furthermore, the type or resolution of the perturbation greatly influences the quality of visualization results.

CAM-based explanations, e.g. CAM (Zhou et al., 2016), Grad-CAM (Selvaraju et al., 2017), and Grad-CAM++ (Chattopadhay et al., 2018), produce a heat map from a selected intermediate layer by linearly combining its feature activation maps with weights that indicate each feature's importance. For example, Grad-CAM defines the weights as the global average pooling of the corresponding gradient map, computed using back-propagation. Some gradient-free CAMs (Ramaswamy et al., 2020; Wang et al., 2020b;a) adopt the perturbation to generate weights from class score changes.

Although Grad-CAM has been adopted to study adversarial context patches in single-shot object detectors (Saha et al., 2020), the explanations are still category-specific. Petsiuk et al. (2021) describes the reasons that make direct application of existing classifier explanation methods infeasible for object detector, and then proposes D-RISE (Petsiuk et al., 2018), a black-box perturbation-based method. Hence, D-RISE inherits the pros and cons of RISE: high-generalizability due to the black-box nature, but time-consuming and noisy due to the inference procedure. Wu & Song (2019) estimates a latent-part model of each detected instance to explain its high-level semantic structure.

To explore white-box explanations of detectors, we propose ODAM, which uses gradient information to generate importance heat map for instance-specific detector explanations. Based on ODAM, we also propose Odam-Train training to improve the object discrimination of the instance-specific explanations, which is infeasible with the class-specific heat maps and the classification task.

**Advanced NMS** Classic NMS assumes that multiple instances rarely overlap, and thus high IoU (intersection over union) of two bounding boxes indicates duplicate detections. More advanced NMS are proposed to mitigate the overdependence on IoU: SoftNMS (Bodla et al., 2017), AdaptiveNMS (Liu et al., 2019), Visibility Guided NMS (Gählert et al., 2020), RelationNet (Hu et al., 2018), and Gnet (Hosang et al., 2017). These methods either use extra predicted cues, but still assume that high IoU correspond to duplicate detections, or modify the detectors to be more complex. Relying on IoU is not enough in crowded scenes where objects partially occlude each other, and thus their IoUs are naturally large. In these cases, internal information about the predictions is required. FeatureNMS (Salscheider, 2021) encodes features for predictions, and trains their distances between the same object to be smaller than those of different objects. In contrast, we propose Odam-NMS, which uses the correlations between instance-level heat maps and the their box IoUs to remove duplicate proposals of the same object. Compared with Salscheider (2021), our Odam-NMS is more stable and can also be interpreted to explain which objects were detected (i.e., object discrimination).

## 3 METHOD

We propose our ODAM for explaining object detection with instance-level heat maps (Sec 3.1), Odam-Train for improving explanations for object discrimination (Sec. 3.2), and Odam-NMS for NMS in crowd scenarios using these explanations (Sec. 3.3).

### 3.1 ODAM: OBJECT DETECTOR ACTIVATION MAPS

Given an image $\mathcal{I}$, the detector model outputs multiple predictions, with each prediction $p$ consisting of the class score $s_c^{(p)}$ and bounding box $B^{(p)} = (x_1^{(p)}, y_1^{(p)}, x_2^{(p)}, y_2^{(p)})$. Our goal is to generate heat maps to indicate the important regions that have a positive influence on the output of each prediction.

In Grad-CAM (Selvaraju et al., 2017) and its generalization Grad-CAM++ (Chattopadhay et al., 2018), the final score for a particular class $Y_c$ is predicted from the whole image and the algorithm

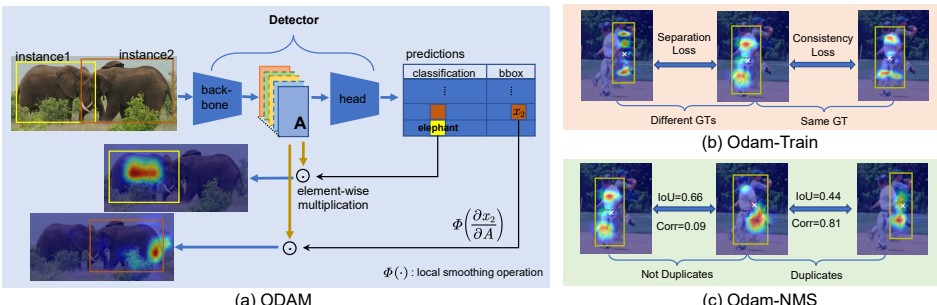

Figure 2: Our proposed framework. (a) ODAM generates instance-specific heat maps for explaining the predictions of an object detector; (b) ODAM-Train uses auxiliary losses to encourage heat maps for predictions on the same object to be consistent, and for different objects to be distinct; (c) ODAM-NMS uses the box IoU and the normalized correlation between heat maps to determine the duplicate detections. The bounding box shows the detection proposal corresponding to the heat map.

ignores distinguishing object instances within. Their explanation starts from the assumption that the score to be interpreted can be written as a linear combination of its global pooled last convolutional layer feature maps $\{A_k\}_k$, $Y_c = \sum_k w_k^c \sum_{ij} A_{ijk} = \sum_{ij} \sum_k w_k^c A_{ijk}$, where $A_{ijk}$ indexes location $(i, j)$ of $A_k$. Thus, the class-specific heat map $H_{ij}^c = \sum_k w_k^c A_{ijk}$ summarizes the feature maps with $w_k^c$ capturing the importance of the k-th feature. To obtain the importance of each feature channel, Grad-CAM estimates $w_k^c$ by global average pooling the gradient map $\partial Y_c / \partial A_k$, while Grad-CAM++ uses a weighted global pooling. However, both methods are limited to class-specific explanations, due to computing a channel-wise importance, which ignores the spatial discrimination information that is essential for interpreting different object instances.

Based on the above analysis, we assume that any predicted object attribute scalar $Y^{(p)}$ of a particular instance $p$ can be written as a linear element-wise weighted combination of the feature map, and then the instance-specific heat map $H_{ij}^{(p)}$ can be produced by summarizing the feature maps with weight $w_{ijk}^{(p)}$ that captures the importance of *each pixel and each channel*,

$$Y^{(p)} = \sum_k \sum_{ij} w_{ijk}^{(p)} A_{ijk}, \quad H_{ij}^{(p)} = \sum_k w_{ijk}^{(p)} A_{ijk}. \quad (1)$$

Previous gradient-based works (Simonyan et al., 2013; Springenberg et al., 2014; Selvaraju et al., 2017; Chattopadhay et al., 2018) have shown that the partial derivative w.r.t. $A_{ijk}$ can reflect the influence of the $k$-th feature at $(i, j)$ on the final output. However, such pixel-wise gradient maps are typically noisy, as visualized in (Simonyan et al., 2013). Thus, we set the importance weight map $w_k^{(p)}$ according to the gradient map $\partial Y^{(p)} / \partial A^k$ after a local smoothing operation $\Phi$, and the corresponding heat map for scalar output $Y^{(p)}$ is obtained through a pixel-weighted mechanism:

$$w_k^{(p)} = \Phi\left(\frac{\partial Y^{(p)}}{\partial A^k}\right), \quad H^{(p)} = \text{ReLU}\left(\sum_k w_k^{(p)} \circ A^k\right), \quad (2)$$

where $\circ$ is element-wise multiplication. Here, local pooling is utilized on the gradient map to produce a smooth weight map, while maintaining the important spatial discrimination information. We adopt a Gaussian kernel for $\Phi$ in the experiments, and adaptively decide the size of the kernel based on the size of predicted object in the feature map. Fig. 2a shows our ODAM framework.

When the scalar output $Y^{(p)}$ is a class score, ODAM highlights the important feature regions used by the detector to classify the instance $p$. Note that $Y^{(p)}$ could be any differentiable attribute of the predictions. For example, in our experiments we also examine the heat maps related to the predicted coordinates of the regressed bounding box (see Fig. 3).

Our work is the first analysis and successful attempt to use a white-box method to generate instance-level explanations for object detector predictions, rather than only class-level explanations. using PyTorch, gradients of any scalar targets w.r.t. intermediate features can be calculated automatically (Paszke et al., 2017). In this way, generating an ODAM explanation for one prediction takes about 2ms, which is much faster than the perturbation-based D-RISE, where using 5000 masks requires $\sim$3 minutes to process one image with FCOS. In Secs. 4.1 and 4.2, we qualitatively and quantitatively verify that our ODAM can effectively explain the predicted results for each detection proposal.

## 3.2 ODAM-TRAIN

Since the instance-specific heat maps may still "leak" onto other neighboring objects, especially in crowded scenes (e.g., see Fig. 4a), we propose a training method Odam-Train for improving the heat maps for object discrimination, to better explain which object was being detected. In order to focus the detector to be better localized on a specific object area, and not overlapped with other objects, Odam-Train encourages similar attention for different predictions of the same object, and separate attentions for different objects (see Fig. 2b). Specifically, we propose a heat-map consistency loss $L_{\text{con}}$ and a separation loss $L_{\text{sep}}$ as auxiliary losses during detector training. Using the predicted confidence scores as explanation targets, heat maps are first calculated by ODAM for all the positive proposals, and then resized to the same size, and vectorized. The ODAM vectors are then organized by ground-truth (GT) object, where $\mathcal{P}^{(p)} = \{H_n^{(p)}\}_n$ is the set of ODAM vectors for positive predictions of the $p$-th GT object. For each GT object, the best prediction $H_{\text{best}}^{(p)}$ is selected from $\mathcal{P}^{(p)}$ that has the highest IoU with the GT box. The *consistency and separation losses* are defined as:

$$L_{\text{con}} = \sum_{p \in GT} \sum_{n \in \mathcal{P}^{(p)}} - \log \cos(H_{\text{best}}^{(p)}, H_n^{(p)}), \; L_{\text{sep}} = \sum_{p \in GT} \sum_{m \notin \mathcal{P}^{(p)}} - \log \left(1 - \cos(H_{\text{best}}^{(p)}, H_m^{(\neg p)})\right), \quad (3)$$

where $H_m^{(\neg p)}$ are ODAM vectors for proposals not corresponding to the $n$-th GT object. The loss for detector training is $L = L_{\text{detector}} + (L_{\text{con}} + L_{\text{sep}})/N$, where $N$ is the total number of ODAM vector pairs during the loss calculation.

## 3.3 ODAM-NMS

In object detection, duplicated detections are removed in post-processing using NMS, which is based on the assumption: two bounding boxes (bboxes) that are overlapped (high IoU) are likely to be duplicated, and the bbox with lower-score (less confidence) should be removed. In particular, for classical NMS, the predictions in list $P$ are sorted by their confidence scores, then each prediction $p \in P$ is compared to the currently selected detections $d \in D$. If the IoU between $p$ and any $d \in D$ is larger than a threshold $T_{iou}$, $p$ is considered a duplicate and discarded. Otherwise $p$ is added to $D$. The classical NMS has shortcomings in crowded scenes because the key assumption does not hold when objects are partially occluded by neighboring objects.

We propose Odam-NMS to mitigate this problem based on an observation that with Odam-Train, the ODAM heat maps for different objects can be distinctive, even though their bboxes are heavily overlapped (see the left and center heat maps in Fig. 2c). Meanwhile, even if the IoU of two predicted bboxes is small, their explanations may be similar indicating the same object instance is detected. For example, in Fig. 2c, the center and right predictions have low IoU but have heat maps with high correlation. In other words, ODAM shows which object the model looked at to make the prediction, which can intuitively assist NMS to better identify duplicate object predictions.

After the inference stage of the detector, we use ODAM to generate heat maps for each prediction with the predicted confidence score. All the heat maps are resized to the same size with a short edge length of 50, then vectorized. Normalized correlation is calculated between each pair of vectors to represent the probability that the two predictions correspond to the same object. Odam-NMS uses both the IoUs and heat map correlations between $p$ and $d \in D$ when considering whether a prediction should be removed or kept. If the IoU is large ($IoU \geq T_{iou}$) and the correlation is very small ($corr \leq T^l$), then $p$ is not a duplicate; If the IoU is small ($IoU < T_{iou}$) and the correlation is very large ($corr > T^h$), then $p$ is a duplicate. Through these two conditions, Odam-NMS keeps more proposals in the high-IoU range for detecting highly-overlapped crowded objects, and removes more proposals in the low-IoU range for reducing duplicates. The pseudo code is in the App. A.1.

## 4 EXPERIMENTS

In this section we conduct experiments on ODAM to: 1) evaluate its visual explanations qualitatively and quantitatively, and compare with the current state-of-the-art method D-RISE (Petsiuk et al., 2021); 2) evaluate the proposed Odam-NMS on crowded scenes. We mainly conduct the experiments with one-stage detector FCOS (Tian et al., 2019) and two-stage detector Faster R-CNN, (Ren et al., 2015) using ResNet-50 (He et al., 2016) as the backbone and FPN (Lin et al., 2017a) as the neck. Two datasets are adopted for evaluation: MS COCO (Lin et al., 2014), a standard object detection dataset, and CrowdHuman (Shao et al., 2018), containing scenes with heavily overlapped objects. Experiments are performed using PyTorch and an RTX 3090 GPU. The training and testing hyperparameters are the same as those of the baseline detectors.

Figure 3: Heat map explanations of two instances (person and dog) computed from different detectors with ResNet50 and pyramid vision transformer (PVT) (Wang et al., 2021) as backbones. (left) The heat maps explain important regions for each predicted attribute (class score $s_c$ and bbox coordinates $x_1, y_1, x_2, y_2$) from FCOS. In FCOS, the bbox coordinates are relative to the anchor center, with positive values indicating a larger box, and thus the highlighted regions for the bbox coordinates are important for expanding the bbox. (right) The combined heat maps for the entire predictions for one-stage RetinaNet, FCOS, the two-stage Faster R-CNN, and transformer-based DETR (Carion et al., 2020). Features from the last stage of ResNet50 are used to explain DETR because there is no detector neck, while features from Feature Pyramid Network (FPN) (Lin et al., 2017a), the detector neck, are used for other methods.

## 4.1 QUALITATIVE EVALUATION OF VISUAL EXPLANATIONS

**Evaluation via visualization** In order to verify the interpretability of visualizations, we generate visual explanations for different prediction attributes (class score, bbox regression values) of two specific instances using various detector architectures with the two types of backbones. To obtain a holistic view of the explanations generated by different models, we compute a combined heat map based on element-wise maximum of heat maps for the predicted class and bbox regression, $H_{\text{comb}} = \max(H_{\text{class}}, H_{x_1}, H_{y_1}, H_{x_2}, H_{y_2})$.

Here we adopt the original baseline detector models (Odam-Train results are visualized in the next experiment). A set of example results are presented in Fig. 3 (left), and we have the following observations from examining many such examples (see App. A.5.5): 1) When predicting the object class, the model attends to the central areas of the object; 2) when regressing the bbox, the model focuses on the extent of the object, (see App. A.5.6 for more examples); 3) For the same target, models from different detectors show attention on different regions, even though they all detect the same instance with a high confidence. Thus, developers can have an intuitive judgment about the model through explanation maps.

From the combination maps (Fig. 3 right), which consider both classification and bbox regression, the right arm of the person and the head of the dog are commonly important for the detectors, although the heat maps look different. More examples are in App. A.5.7.

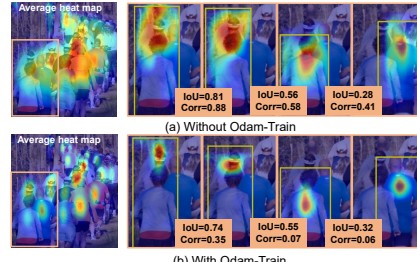

We further compare the visualizations of Grad-CAM, D-RISE and our ODAM (w/o and w/ Odam-Train) in Fig. 1. The results are all generated with FCOS with FPN, and both Grad-CAM and ODAM use class score targets, while D-RISE uses the same mask settings as Petsiuk et al. (2021) to find the attention area of predictions. Our ODAM demonstrates a strong ability of generating clear and distinct instance-specific heat maps. In contrast, Grad-CAM is class-specific (highlighting all the people for each detected person), and D-RISE contains "speckles" in the background due to the random masking mechanism. See App. A.5.4 for more examples.

Figure 4: Comparison of heat maps from FCOS without and with Odam-Train. (left) The average heat map over the high-quality predictions with confidence score over 0.1. (right) Instance-specific heat maps of some predictions on different objects, with the IoU and correlation between each pair of predictions displayed in the middle.

**Visualization w/ and w/o Odam-Train** Next we compare the heat maps from ODAM using the baseline detector trained with and without Odam-Train. The heat maps for each proposal are computed by ODAM with its confidence score as the explanation target. Although the original heat maps without Odam-Train (Fig. 4a) can locate the object well, the attention is spread to its overlapping neighbors, which makes the correlations between them relatively high. Using the consistency and separation losses in (3), Odam-Train yields well-localized heat maps for the same object and distinctive heat maps for different objects, which better shows which object was being detected, improving

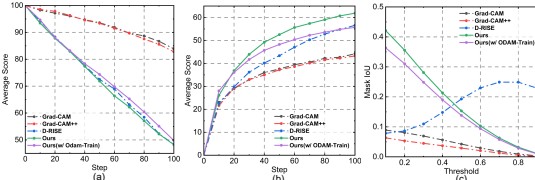

Figure 5: Average prediction score vs. (a) Deletion steps and (b) Insertion steps. (c) The IoU between the ground truth object mask and thresholded explanation heat map.

Table 1: Evaluation of faithfulness: AUC for Deletion, Insertion and Visual Explanation Accuracy (VEA) curves in Fig. 5.

| Method | Del.↓ | Ins.↑ | VEA ↑ |
|---|---|---|---|
| Grad-CAM | 92.79 | 36.78 | 0.039 |
| Grad-CAM++ | 92.52 | 36.18 | 0.027 |
| D-RISE | 73.35 | 43.35 | 0.157 |
| ODAM | **72.68** | **50.33** | **0.163** |
| w/ Odam-Train | 74.45 | 46.66 | 0.143 |

object discrimination. As seen in Fig. 4b, different people in the crowd can be separated by their low heat-map correlation, even when they have high IoU.

## 4.2 QUANTITATIVE EVALUATION OF VISUAL EXPLANATIONS

We evaluate faithfulness using Deletion, Insertion (Samek et al., 2016; Chattopadhay et al., 2018; Wang et al., 2020b;a; Petsiuk et al., 2021) and visual explanation accuracy (Oramas Mogrovejo et al., 2019), and also evaluate on the user trust of the produced heat maps. For evaluating localization, we adopt Pointing games (Zhang et al., 2018). Meanwhile, we propose an object discrimination index (ODI) for measuring the interpretation ability of object discrimination. For comparison, we implement D-RISE using 5000 masks with resolution of 16×16 as in (Petsiuk et al., 2021). FCOS is used as baseline model and the features from FPN are adopted to generate heat maps. The best matched predictions of the well-detected objects ($IoU > 0.9$) in the evaluation dataset are interpreted by each explanation method. The confidence scores of predictions are used as the explanation targets in ODAM, Grad-CAM and Grad-CAM++. Besides the MS COCO val set, results of the Pointing game and ODI are also reported on CrowdHuman validation sets.

**Deletion and Insertion** A faithful heat map should highlight the important context on the image. Deletion replaces input image pixels by random values step-by-step using the ordering of the heatmap (most important first), then measures the amount of the predicted confidence score drops (in percentage). Insertion is the reverse operation of deletion, and adds image pixels to an empty image in each step (based on heatmap importance) and records the average score increase. The average prediction score curves are presented in Fig. 5(a-b), and Tab. 1 reports the area under the curve (AUC). Lower Deletion AUC means steeper drops in score, while higher Insertion AUC means larger increase in score with each step. Our methods have the fastest performance drop and largest performance increase for Deletion and Insertion, which shows that the regions highlighted in our heat maps have larger effects on the detector predictions, as compared to other methods. Faithfulness metrics when using Odam-Train are slightly worse than without it, but still better than previous works on Insertion and VEA. Note that instance-specific explanations (ours and D-RISE) significantly surpass the classification explanation methods (Grad-CAM, Grad-CAM++).

**Visual explanation accuracy (VEA)** VEA measures the IoU between the GT and the explanation heat map thresholded at different values. We use the MS COCO GT object masks for this evaluation, and results are presented in Fig.5(c). Our method obtains the highest IoU when the threshold is small ($T < 0.4$), and the IoUs decrease as the threshold increases. This demonstrates that our heat map energy is almost all inside the object (consistent with energy PG results in Tab. 4). In contrast, the low IoUs of the previous methods at a small threshold indicate that their heatmaps contains significant amounts of energy outside the object mask. For Grad-CAM/Grad-CAM++, the IoUs decrease when the threshold increases, which suggests that there are more large heat map values outside the object mask than inside the mask (they are not object-instance specific). For D-RISE, the IoU increases as the threshold increases, which indicates that the heat map values inside the object mask are larger than those outside the mask. Overall, our method has better IoU (0.421 at T=0.1) compared to D-RISE (0.249 at T=0.7), which is also verified by the VEA AUC in Tab. 1.

**User Trust** The interpretability of visual explanations is evaluated through a human trust test. Heat maps are generated by D-RISE, Grad-CAM, Grad-CAM++, and ODAM for 120 correctly-detected objects out of 80 classes from the MSCOCO val set (1-2 instances for each class). For each object, users are asked to rank the maps from the four methods by the order of which map gives more reasonable insight about how the target object was detected. We collect 10 responses for each object from a total number of 40 users (30 objects per user), totaling 1200 responses. The results are presented in Tab. 2. ODAM is ranked 1st place 53.8% of trials and 2nd place 35.4%, which is significantly better than D-RISE ($\chi^2$ test, p<0.001). Overall, ODAM has significantly better average

Table 2: User trust results: (left) percentage of rankings for each method; (right) average rank (AR).

| Method | 1st | 2nd | 3rd | 4th | AR |
|---|---|---|---|---|---|
| Grad-CAM | 3.9 | 12.9 | 30.5 | 52.7 | 3.3 |
| Grad-CAM++ | 7.3 | 22.2 | 43.1 | 27.5 | 2.9 |
| D-RISE | 35.1 | 29.5 | 17.5 | 17.9 | 2.2 |
| ODAM | **53.8** | 35.4 | 8.9 | 1.9 | **1.6** |

Table 3: (a) Comparison of Object Discrimination Index (%) using GT bbox or mask. (b) User-study on object discrimination. CH is CrowdHuman dataset.

| (a) | MS COCO | | CH | (b) CH | |
|---|---|---|---|---|---|
| | box ↓ | mask ↓ | box ↓ | Acc ↑ | Conf ↑ |
| Grad-CAM | 77.0 | 72.7 | 91.4 | 14.2 | 1.7 |
| Grad-CAM++ | 77.3 | 73.2 | 92.0 | 18.8 | 1.5 |
| D-RISE | 71.0 | 66.3 | 95.3 | 60.7 | 2.4 |
| ODAM | *34.8* | *19.5* | *56.9* | *94.0* | *4.6* |
| w/ Odam-Train | **34.1** | **18.7** | **51.3** | **94.7** | **4.8** |

Table 4: Comparison of Pointing Game (PG) accuracy with ground-truth bounding boxes (b) or segmentation masks (m), energy-based PG with box or mask, and Heat Map Compactness (Comp.).

| | MS COCO | | | | | CrowdHuman | | |
|---|---|---|---|---|---|---|---|---|
| | PG(b)↑ | PG(m)↑ | enPG(b) ↑ | enPG(m) ↑ | Comp.↓ | PG(b)↑ | enPG(b) ↑ | Comp.↓ |
| Grad-CAM | 26.7 | 22.5 | 20.7 | 15.0 | 4.34 | 15.7 | 9.7 | 3.99 |
| Grad-CAM++ | 26.6 | 20.2 | 20.0 | 14.8 | 4.91 | 15.4 | 11.4 | 3.84 |
| D-RISE | 82.6 | 68.0 | 17.4 | 12.0 | 5.17 | 1.5 | 1.7 | 3.53 |
| Ours | *91.9* | *82.6* | *73.1* | *57.1* | *1.36* | *95.5* | *79.5* | *1.04* |
| Ours w/ Odam-Train | **93.3** | **83.9** | **79.6** | **63.9** | **1.32** | **97.3** | **83.9** | **0.91** |

rank of 1.6 compared to others (Wilcoxon signed-rank test, p<0.001). The significantly higher human trust of ODAM demonstrate its superior instance-level interpretability for object detection.

From another aspect, previous studies evaluated trust between humans and machine learning systems by seeing if better models had better explanation maps according to humans. Following Selvaraju et al. (2017); Petsiuk et al. (2021), ODAM maps are generated for 120 objects that are correctly detected by two FCOS-ResNet50 models with different performance (36.6% mAP and 42.3% mAP). Excluding the samples where users thought the explanations were similar quality, the better model (42.3% mAP) received more responses that its explanations were more trustworthy (38.2% vs. 28.6%). More details are provided in the Appendix A.2.

**Pointing Game (PG)**  To quantitatively evaluate the localization ability, we report results of the PG metric. For PG, the maximum point in the instance-level heat map is extracted and a hit is scored if the point lies within the GT object region (either bbox or instance mask). Then the PG accuracy is measured by averaging over the test objects. Since PG only considers the maximum point, but not the spread of the heat map, we also adopt the energy-based PG (Wang et al., 2020b), which calculates the proportion of heat map energy within the GT object bbox or mask (versus the whole map). Finally, to show the *compactness* of the heat map, we calculate the weighted standard deviation of heat-map pixels, relative to the maximum point: $Comp. = \left( \frac{1}{\sum_x S_x} \sum_x \frac{S_x ||x - \hat{x}||^2}{\frac{1}{4}(h^2 + w^2)} \right)^{\frac{1}{2}}$, where $S_x$ is the heat map value at location $x$, $\hat{x}$ is the maximum point of the heat map, and $(w, h)$ are the width and height of the GT box. The denominator normalizes the distance w.r.t. the object size. Smaller compactness values indicate the heat map is more concentrated around its maximum.

Both Grad-CAM and Grad-CAM++ perform poorly on both datasets since they generate class-specific heat maps. Our ODAM yields significant improvements over D-RISE on all the metrics. Specifically, D-RISE cannot work well on CrowdHuman, which only contains one object category. Since D-RISE uses the similarities between predictions of masked images and the original image to decide the mask weights, for datasets with few object categories, the predicted class probabilities provide less useful information when calculating the weights. Finally, using Odam-Train further improves the localization quality by generating heat maps that are well-localized on the objects.

**Object Discrimination**  To evaluate the object discrimination ability of the heat maps, we propose the object discrimination index (ODI), which measures the amount of heat map energy that leaks to other objects. Specifically, for a given target object, the ODI is the proportion of heat map energy inside all other objects' bboxes (or segmentation masks) w.r.t. the total heat map energy inside all objects in the image (i.e., ignoring background regions). The averaged ODIs are presented in Tab. 3a. Compared with others, ODAM consistently shows the least energy leaking out to other objects. Note that when using the tighter GT mask on MSCOCO, ODAM obtains the largest proportion of decrease, which indicates that the heat map can better focus on the explained target, even if its bbox overlaps with other objects. Using Odam-Train significantly improves object discrimination in the crowded scenes of CrowdHuman, although there is a tradeoff with faithfulness (see Tab. 1). Finally, we conduct a user study on object discrimination (details in App. A.3). The results in Tab. 3b show

Table 5: Comparisons of NMS strategies on CrowdHuman validation set. All models are trained with the same baseline implementation. The timing is for the whole pipeline: detector and heat map inference, and NMS.

| | FCOS | | | | | Faster RCNN | | | | |
|---|---|---|---|---|---|---|---|---|---|---|
| | AP↑ | JI↑ | MR↓ | Recall | time (s/img) | AP↑ | JI↑ | MR↓ | Recall | time (s/img) |
| NMS | 87.8 | 78.4 | 45.5 | 93.2 | 0.114 | 86.9 | 79.5 | 43.2 | 90.3 | 0.092 |
| Soft-NMS | 80.8 | 74.9 | 89.0 | 93.0 | 0.47 | 76.5 | 61.9 | 84.8 | 92.3 | 0.284 |
| FeatureNMS | 89.3 | 78.1 | 45.6 | 95.4 | 0.145 | 82.0 | 65.7 | 68.8 | **94.9** | 0.120 |
| Odam-NMS (ours) | **89.3** | **81.1** | **44.5** | **95.5** | 0.178 | **88.1** | **80.5** | **42.8** | 91.5 | 0.140 |

that users are more accurate and more confident at object discrimination using ODAM compared to previous works ($\chi^2$ test, p<.0001; t-test, p<.0001), and that using Odam-Train further improves user confidence (t-test, p=.02). These results are consistent with the quantitative evalution with ODI.

### 4.3 Evaluating Odam-NMS

We evaluate Odam-NMS with Odam-Train for improving detection in crowded scenes. To evaluate the performance of NMS on heavily overlapped situations, we adopt the CrowdHuman dataset, which contains an average 22.6 objects per image (2.4 overlapped objects). We compare Odam-NMS with NMS, SoftNMS, and FeatureNMS, using both FCOS and Faster RCNN. The IoU threshold is set to $T_{iou} = 0.5$ for both NMS and our method. Soft-NMS uses Gaussian decay with $\sigma = 0.5$ and final threshold of 0.05. For FeatureNMS, the IoU range is set to $(0.9, 0.1)$ following Salscheider (2021). The aspect ratios of the anchors in Faster R-CNN are set to $H : W = \{1, 2, 3\}$ based on the dataset, and other parameters are the same as in the baselines. Training runs for 30 epochs. We use $T^h = 0.8$, $T^l = 0.2$ based on the ablation study in the App. A.4.1. We adopt three evaluation criteria: *Average Precision* ($AP_{50}$); *Log-Average Missing Rate* (MR), commonly used in pedestrian detection, which is sensitive to false positives (FPs); *Jaccard Index* (JI). See Chu et al. (2020) for details. Smaller MR indicates better results, while larger $AP_{50}$ and JI are better.

Tab. 5 shows the results. Soft-NMS performs poorly in crowd scenes, generating many false positives in high-score region (high MR) with a long processing time. For FCOS, AP performance of FeatureNMS is much higher than NMS, while JI and MR are similar. However for Faster RCNN, although FeatureNMS obtains a high recall, the others are worse than NMS, indicating that the feature embeddings trained with the cropped features in two-stage detectors are not distinctive enough, and there are many false positives in detections. The learned embeddings in FeatureNMS have no explicit meaning except the relative distance between each pair, while Odam-NMS directly uses heat maps that offer explanations of the detector model. With the default IoU threshold, our Odam-NMS achieves better JI and MR than NMS and FeatureNMS for both detectors (FCOS and Faster RCNN). Meanwhile, Odam-NMS also achieves the best AP with Faster RCNN. The limitation of Odam-NMS is that generating heat maps for dense predictions takes slightly longer. Overall, these results verify the object discrimination interpretation ability of ODAM with Odam-Train and demonstrate that the instance-level explanation for predictions can help improve NMS in crowd scenes.

## 5 Conclusion

In this paper, we propose ODAM, a white-box gradient-based instance-level visualized explanation technique for interpreting the predictions of object detectors. ODAM can produce instance-specific heat maps for any prediction attribute, including object class and bounding box coordinates, to show the important regions that the model uses to make its prediction. Our method is general and applicable to one- and two-stage and transformer-based detectors with different detector backbones and heads. Qualitative and quantitative evaluations demonstrate the advantages of our method compared with the class-specific (Grad-CAM) or black-box works (D-RISE). We also propose a training scheme, OdamTrain, which trains the detector to produce more consistent explanations for the same detected object, and distinct explanations for different objects, thus improving object discrimination. Experiments show that Odam-Train is effective at improving localization in the explanation map and object discrimination. Finally, we propose Odam-NMS, which utilizes the explainability of ODAM to identify and remove duplicate predictions in crowd scenarios. Experiments on different detectors confirm its effectiveness, and further verify the interpretability provided by ODAM for both object specification and object discrimination. Finally, we note that there appears a tradeoff between faithfulness and object discrimination in the explanation maps, since high object discrimination implies not using context information. Future work will consider how to manage this tradeoff.

ACKNOWLEDGMENTS

This work was supported by a grant from the Research Grants Council of the Hong Kong Special Administrative Region, China (Project No. CityU 11215820).

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

# A APPENDIX

## A.1 IMPLEMENTATION DETAILS: ODAM-TRAIN AND ODAM-NMS

For Odam-Train on MSCOCO, the detector training pipeline is totally the same as the baseline (Tian et al., 2019; Ren et al., 2015), which uses SGD as the optimizer running for 12 epochs with batch-size 16, learning rate 0.2 for two-stage Faster R-CNN and learning rate 0.1 for FCOS. For training on CrowdHuman, the aspect ratios of the anchors in Faster R-CNN are set to $H : W = \{1, 2, 3\} : 1$ since the dataset contains people, and training runs for 30 epochs. Other parameters are the same as in training on MSCOCO.

Since there are many predictions from the object detector, calculating gradients of each prediction w.r.t. the feature maps one-by-one will incur an unacceptably long time cost for ODAM-Train and ODAM-NMS. To enable efficient ODAM-Train and ODAM-NMS, we adopt the RoI-pool features in the two-stage detector as $A_k$, since the gradients w.r.t. this layer for all predictions can be computed in a batch with the "autograd" function in PyTorch. As for one-stage detectors, the output features from the Feature Pyramid Network (FPN) Lin et al. (2017a) are adopted, and their gradients are computed in a batch through expansion of the gradient calculations Liu & Chan (2022) into a "reversed" detector head. This substantially improves the efficiency of ODAM-Train and ODAM-NMS (see Tab. 5 for the time cost per image).

The pseudo-code for Odam-NMS is presented in Algorithm 1.

---

**Algorithm 1** Odam-NMS: predictions are removed or kept based on both the IoU and the correlation between ODAM heat maps.

---

```
P ← GetPredictions(imageI)
P ← SORT(P)
D ← ∅
while P ≠ ∅ do
    p ← POP(P)
    isDuplicate ← false
    for d ∈ D do
        iou ← GetIoU(p, d)
        corr ← NormCorrelation(S_{y_c}^{(p)}, S_{y_c}^{(d)})
        if iou ≥ T_{iou} and corr > T^l then
            isDuplicate ← true
        else if iou < T_{iou} and corr > T^h then
            isDuplicate ← true
        end if
    end for
    if ¬isDuplicate then
        PUSH(p, D)
    end if
end while
```

---

## A.2   USER TRUST ON EXPLANATION FAITHFULNESS EVALUATION

We present the form of the questionnares and some samples that we used in the two user trust tests in Fig. 6.

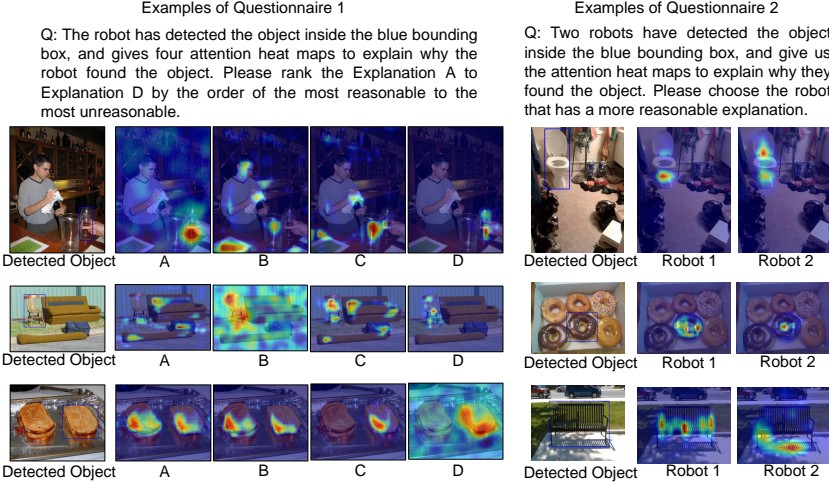

Figure 6: In the first questionnaire, the human needs to rank the heat maps from four explanation methods, including D-RISE, Grad-CAM, Grad-CAM++ and ODAM (ours). In the second questionnaire, the human is asked to choose the more reasonable heat map from the two explanations, which are generated from models of different performance (36.6% mAP and 42.3% mAP). The labels for options are assigned randomly for each question.

## A.3   USER TRUST ON OBJECT DISCRIMINATION EVALUATION

Since the user trust study in Sec. 4.2 only evaluate explanations from the faithfulness aspect, which shows how reasonable the heat maps explain the predictions, we further conduct a user study on object discrimination ability. In this test, users are asked to draw a bounding box on the image, pointing out which object they think the AI has detected, based on the shown heat map. Meanwhile, the users also need to provide their confidence level from most uncertain (1) to most certain (5) when making the choice. For each method (D-RISE, Grad-CAM, Grad-CAM++, ODAM and ODAM with Odam-Train), 150 samples are sent out to 10 users with one user answering 15 images. Since the purpose is to test whether the heat maps can effectively show which object was detected, especially in the crowded scene, we choose the samples from the CrowdHuman validation set. Since users will have different ways to draw the box around the object, a separate marker manually inspected the user's boxes to see if they align with the GT object box in order to determine correctness. During this process, the marker does not know which explanation method was used for each image.

Tab. 6 presents the number of examples (in percentage) under each confidence level and the accuracy of users' decisions (the ratio of users' correct choices) based on heat maps of each method. The results show that the user can obtain a much higher accuracy ($\chi^2$ test, p<.0001) and confidence (t-test, p<.0001) with heat maps from both ODAM and ODAM with Odam-Train. Furthermore, using Odam-Train further improves the average user confidence (t-test, p=0.024). Using Odam-Train, users have higher confidence (83.99% vs. 70.68% at most certain) about their decision, which demonstrates our method is superior on object discrimination ability, especially with the model after Odam-Train. Some incorrect and correct user's choices are displayed in Fig. 7.

Table 6: Results of user study on object discrimination: (top) the percentage of confidence levels of user responses; (bottom) the average confidence, and the accuracy for correct object discrimination.

| Confidence | Grad-CAM | Grad-CAM++ | D-RISE | ODAM | ODAM w/ Odam-Train |
|---|---|---|---|---|---|
| 1 (least) | 53.38 | 63.76 | 20.67 | 0 | 0 |
| 2 | 30.41 | 24.16 | 36.67 | 0.67 | 1.35 |
| 3 | 10.14 | 6.71 | 26.01 | 7.33 | 3.33 |
| 4 | 5.41 | 4.70 | 11.98 | 21.34 | 11.33 |
| 5 (most) | 0.68 | 0.67 | 4.68 | 70.68 | 83.99 |
| avg. conf. | 1.70 | 1.54 | 2.43 | 4.62 | 4.78 |
| accuracy | 14.19 | 18.79 | 60.67 | 94.00 | 94.67 |

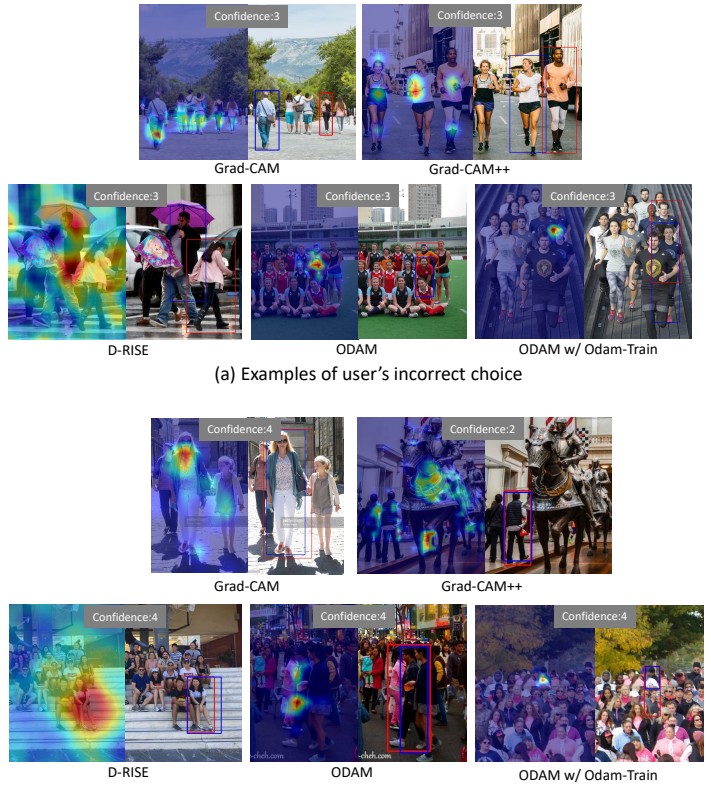

(a) Examples of user's incorrect choice

(b) Examples of user's correct choice

Figure 7: Examples of (a) user's incorrect choices and (b) user's correct choices with heat maps of Grad-CAM, Grad-CAM++, D-RISE, ODAM and ODAM with Odam-Train, respectively. In the user trust test on object discrimination, users are asked to draw the bounding box of the object which was detected based on the given heat map. Blue boxes are those drawn by users, while red boxes are those of the ground truth objects. Note that in CrowdHuman, the ground-truth boxes are for the full person, even when the person is partially occluded. The user's confidences when choosing the objects are also displayed.

Fig. 8 shows a more detailed view of the histogram of users' correctness and confidences compared with the ODIs (object discrimination index, proposed in Sec. 4.2) of the tested samples. Higher ODI means more heat map energy is leaking out to other objects, while less energy is in the explained target object. Except for ODAM (w/ or w/o Odam-Train), most heat maps of other methods have high ODIs (larger than 0.6). We can find that users tend to have lower confidence and make incorrect decisions with these high-ODI heat maps from other methods. In contrast, for ODAM (w/ and w/o Odam-Train), users have higher confidence and make correct decisions even for high ODI samples. These results demonstrate that ODAM can improve the user trust in the object discrimination of the detector.

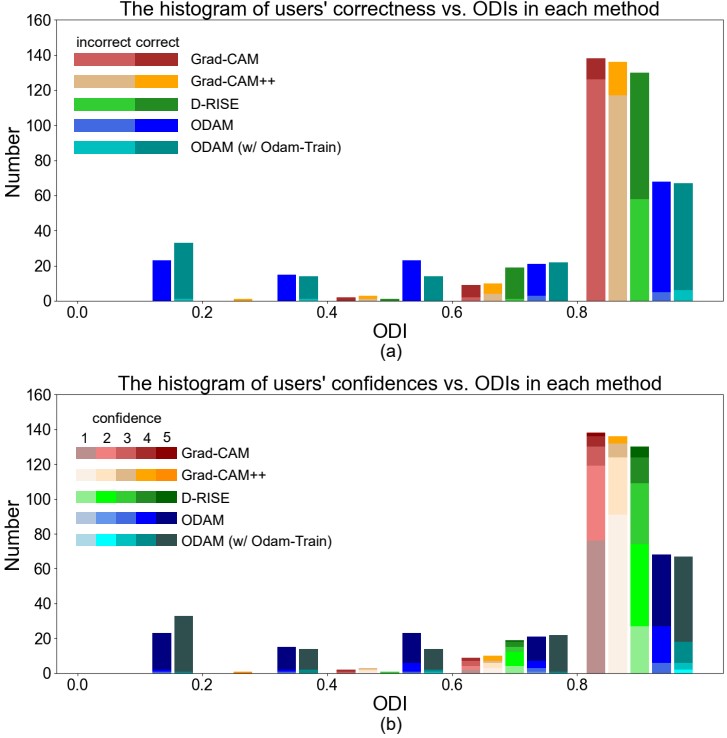

Figure 8: User study on object discrimination: the histogram of users' (a) correctness and (b) confidences vs. the test samples' object discrimination index (ODI).

### A.4 MORE ABLATION STUDIES

#### A.4.1 EFFECTS OF THRESHOLDS IN ODAM-NMS

Figure 9 shows the results comparing various combinations of low and high correlation thresholds, $T^l$ and $T^h$, for Odam-NMS with the FCOS detector. We compare with classical NMS, corresponding to $T^l = 0, T^h = 1$. The results for all the three metrics are improved by increasing $T^l$ to 0.1-0.2, which indicates that some high-IoU crowded objects are successfully kept through their low correlation values. Then with the $T^l$ further increasing, MR and JI starts to get worse, since more false positives are introduced. As for $T^h$, two proposals are regarded as the same object if their correlation exceeds $T^h$, even though their IoU is smaller than the $T_{iou}$. Since using $T^h < 1$ will reduce the recall rate, and $AP_{50}$ is highly depends on the recall, $T^h$ shows no benefit to $AP_{50}$. MR and JI have better performance by decreasing the $T^h$ when $T^l$ is fixed. This indicates that using a $T^h < 1$ can indeed remove low-IoU false positives, although the performance improvement is slight. Based on the results in Fig. 9, we adopt $T^h = 0.8$ and $T^l = 0.2$ in the experiments.

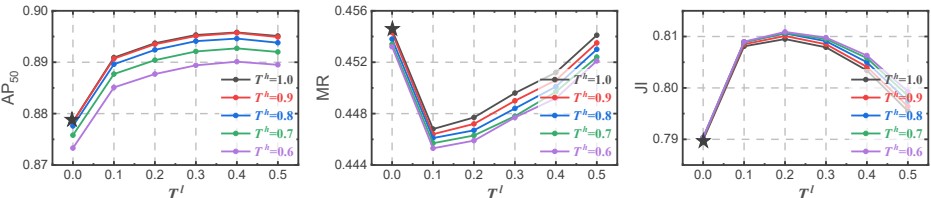

Figure 9: Effects of thresholds $T^l$ and $T^h$ in Odam-NMS. The black star marks the results for classical NMS, in which $T^l = 0$ and $T^h = 1$.

#### A.4.2 ANALYSIS ON RECALLS WITH ODAM-NMS

To verify that more crowded objects can be recalled using Odam-NMS, we compare the recalls of the baseline detectors with or without Odam-NMS for both crowded and uncrowded instances. Since the recall is related to the confidence score threshold, we use the thresholds corresponding to the best JI for each method for fair comparison. Based on (Chu et al., 2020), we define the "crowd" set and "sparse" set, where the ground-truth boxes that overlap with some other ground truth with $IoU > 0.5$ comprise the "crowd" set and the other GT boxes as the "sparse" set. The results are shown in Tab. 7. The recall percentage on the crowd set is much lower than on the sparse set. Using Odam-NMS improves the recall percentage of crowded objects compared to the baseline, e.g., recall is improved by 4% for Faster R-CNN and 13.5% for FCOS.

Table 7: Comparison of recalls on the "crowd" and "sparse" set from CrowdHuman validation set.

| | Ground truth | FasterRCNN | FasterRCNN +ODAM-NMS | Δ | FCOS | FCOS +ODAM-NMS | Δ |
|---|---|---|---|---|---|---|---|
| Total | 99,481 | 79,090 (79.5%) | **80,111 (80.5%)** | **+1%** | 74,946 (75.3%) | **80,650 (81.1%)** | **+5.8%** |
| Sparse | 78,273 | 65,480 (83.6%) | **65,639 (83.8%)** | **+0.2%** | 61,890 (79.0%) | **64,726 (82.7%)** | **+3.7%** |
| Crowd | 21,208 | 13,610 (64.2%) | **14,472 (68.2%)** | **+4%** | 13,056 (61.6%) | **15,924 (75.1%)** | **+13.5%** |

#### A.4.3 ODAM-NMS WITH AND WITHOUT ODAM-TRAIN

In Sec. 4.2, we evaluate Odam-NMS with the model when using Odam-Train. Here we provide the ablation studies evaluating Odam-NMS with and without using Odam-Train. The results are shown in the Tab. 8. When using the classical NMS, the baseline detector model yields similar and comparable results with and without using Odam-Train, which demonstrates that Odam-Train will have little influence on the baseline model performance when improving the explanation ability. This is a desirable property since we hope that providing explanations will not hinder the detector performance. As for Odam-NMS, Odam-Train brings an obvious improvement with the same parameter setting ($T^h = 0.8$ and $T_l = 0.2$) as compared to without Odam-Train. This shows that model can be trained to give more consistent and separated explanations that benefit duplicate detection removal. Furthermore, without Odam-Train, Odam-NMS needs the stricter judgment conditions to make more reasonable decisions, such as increasing $T^h$ and reducing $T_l$, to obtain better results.

Table 9 illustrates the recalls on the "crowd" and "sparse" set when the FCOS model is trained with or without Odam-Train, and uses classical NMS or Odam-NMS in the post-processing. When using

Table 8: Comparisons of NMS performances with and without Odam-Train on the CrowdHuman validation set for both FCOS and Faster RCNN detectors. $T^h$ and $T^l$ are the correlation thresholds in the Algorithm 1. The IoU threshold for all NMS methods is 0.5.

| | Odam-Train | $T^h$ | $T^l$ | FCOS AP↑ | JI↑ | MR↓ | Faster RCNN AP↑ | JI↑ | MR↓ |
|---|---|---|---|---|---|---|---|---|---|
| NMS | | - | - | 87.8 | 78.4 | 45.5 | 86.9 | 79.5 | 43.2 |
| | √ | - | - | 87.5 | 78.9 | 45.6 | 87.0 | 79.3 | 43.8 |
| Odam-NMS | | 0.8 | 0.2 | 87.5 | 78.5 | 54.0 | 88.9 | 78.7 | 44.3 |
| | | 0.9 | 0.1 | 88.6 | 80.4 | 45.5 | **89.0** | 79.8 | 43.7 |
| | √ | 0.8 | 0.2 | **89.3** | **81.1** | **44.5** | 88.1 | **80.5** | **42.8** |

NMS with Odam-Train or Odam-NMS without Odam-Train, the recall rate is improved compared with the classical NMS without Odam-Train. However, the comprehensive metrics (e.g. AP, JI and MR) are not better accordingly (shown in Tab. 8) because the number of false positives is also increased. Specifically, using Odam-NMS with Odam-Train, the recall is best on the crowd set and also obtains superior performance in Tab. 8 by reducing the number of false positives.

Table 9: Comparison of recalls using NMS and Odam-NMS for FCOS detector model with and without Odam-Train on the "crowd" and "sparse" set from CrowdHuman validation set. "Conf." is the confidence threshold used to give the best JI score. For Odam-NMS, the correlation thresholds $T^h$ and $T^l$ are set to 0.8 and 0.2.

| | GT | NMS | | Odam-NMS | |
|---|---|---|---|---|---|
| Odam-Train | - | | √ | | √ |
| Total | 99,481 | 74,946 (75.3%) | 78,097 (78.5%) | **80,697 (81.1%)** | 80,650 (81.1%) |
| Sparse | 78,273 | 61,890 (79.0%) | 64,252 (82.1%) | **65,402 (83.6%)** | 64,726 (82.7%) |
| Crowd | 21,208 | 13,056 (61.6%) | 13,845 (65.3%) | 15,295 (72.1%) | **15,924 (75.1%)** |

In summary, the ablation studies indicate that, since Odam-NMS makes decisions based on heat-map correlations, it relies on good quality explanations, and using Odam-Train is beneficial because it encourages the model to produce consistent and distinctive heat maps on detections of the same or different object.

## A.5 MORE EVALUATIONS OF VISUAL EXPLANATIONS

Here we provide more evaluations about the visual explanations generated by ODAM.

### A.5.1 SANITY CHECKS

Adebayo et al. (2018b) develops sanity checks for saliency methods. Specifically, they compare explanation methods to edge detection techniques, and show that some methods are independent of both the model or training data, but still produce outputs visually similar to those of explanation methods. We adopt the model parameter randomization test proposed in (Adebayo et al., 2018b) to compare the predicted heat maps of the trained model and the untrained model. Fig. 10 shows that the weight randomization results in totally incorrect predictions and corresponding heat maps, which means that our method is related to the predicted instances and relies on the model parameters to produce the explanations.

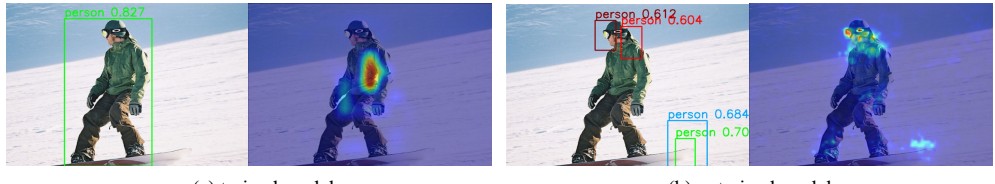

(a) trained model          (b) untrained model

Figure 10: Sanity check. The predictions of "person" and corresponding heat maps generated from trained model and untrained model (with weight randomization).

### A.5.2 ANALYZING ERROR MODES OF DETECTOR

We use ODAM to analyze the error modes of detector. For the high confidence but poorly localized cases, we generate explanations of the wrong predicted extents and compare them with the correct localization results. As shown in Fig. 11, the model highlights that the wrong extents were misled by the leg of the person and the sea horizon.

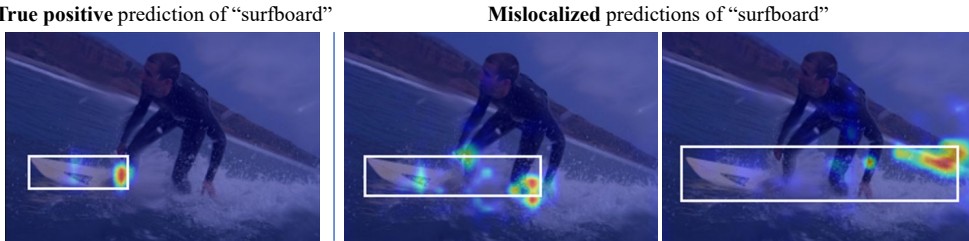

**True positive** prediction of "surfboard"      **Mislocalized** predictions of "surfboard"

Figure 11: Explanations of the predictions of the right extent ($x_2$ in bounding boxes) for different predictions of "surfboard". The heat maps for the mislocalized predictions highlight the visual features that induced to the wrong extents (the leg on the right, and the sea horizon).

To analyze classification decisions of the model, we generate explanations of the class scores. In Fig. 12, the model correctly classifies an instance as "bed" when seeing the cushion of the bed, but also mistakenly predicts "bench" based on a long metal bed frame at the end of the bed. In another example, a person is fixing a wheel on the ground, and two motorcycles are parked nearby. The detector correctly finds the person, but also mistakenly detects a motorcycle on the person, by combining the features from the two motorcycles. This shows a failure mode of the detector, where sometimes the context feature (a person next to unrelated motorcycle parts) may bring negative influence to the detection result.

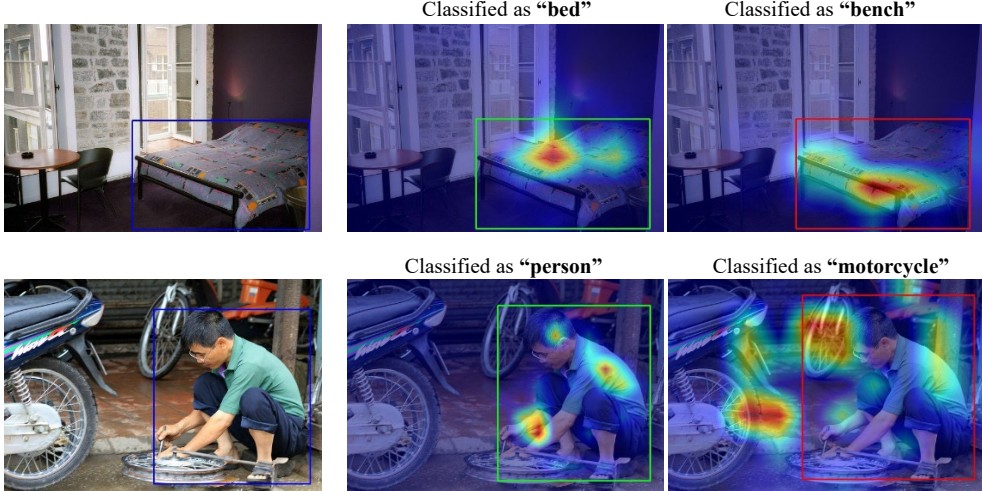

Figure 12: Explanations of the class scores of different predictions. In the first row, the model predicts "bench" when it puts attention on only the frame at the end of the bed. In the second row, the model is negative influenced by the context feature and misclassifies a "motorcycle" on a "person".

### A.5.3 COMPARISON OF EXPLANATIONS FROM DIFFERENT FEATURE MAP LAYERS

In Fig. 13, we visualize and compare the heat maps generated with different feature layers. For the same target, higher-level layers (e.g. FPN and RoI pooling) show more concentrated attention and generate smoother heat maps.

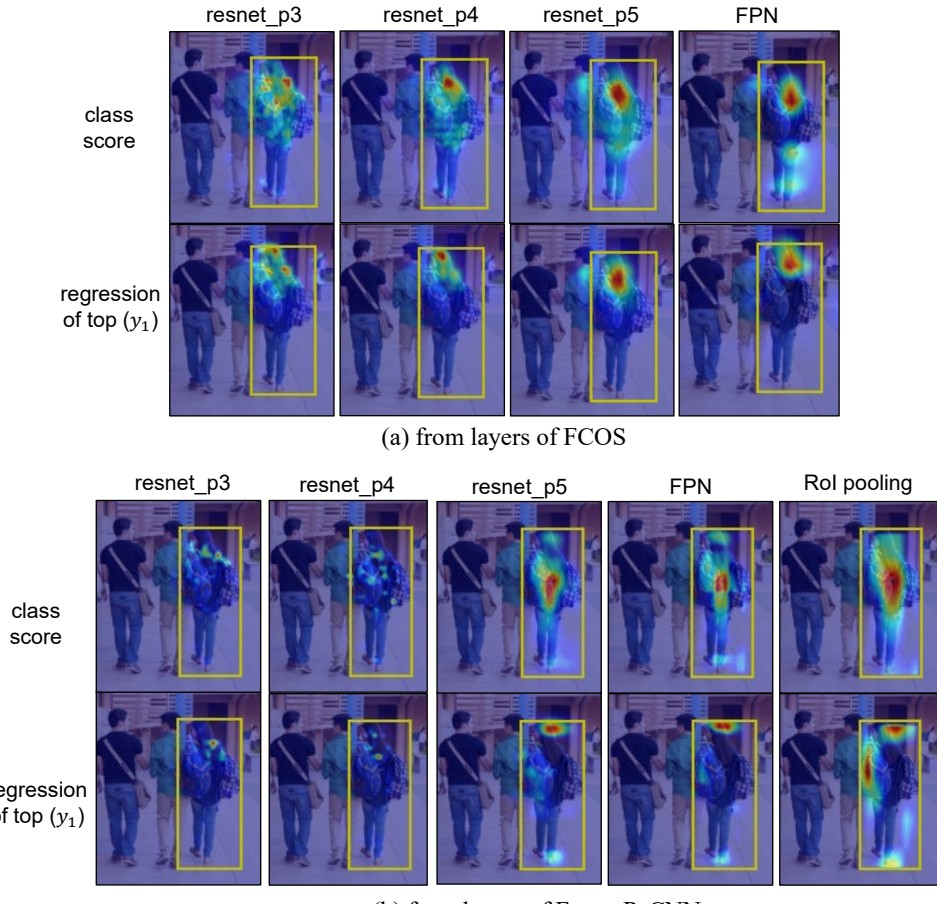

Figure 13: Heat maps computed from different feature maps of the one-stage FCOS and the two-stage Faster R-CNN, when interpreting the class score and the regression of the top extent ($y_1$ in bounding box), respectively. The feature maps $A^{(k)}$ are selected from the output feature maps of the ResNet backbone (resnet_p3-p5), which are also the inputs of Feature Pyramid Network (FPN) with P3-P5 level, the FPN ouput, or the RoI pooling output. Note that the heat map for the RoI Pooling layer is obtained by bilinear interpolating the original map from $7 \times 7$ to the image size.

### A.5.4 MORE COMPARISONS OF DIFFERENT EXPLANATION METHODS

We provide more comparisons of heat maps from D-RISE, Grad-CAM, Grad-CAM++, and our ODAM in Fig. 14.

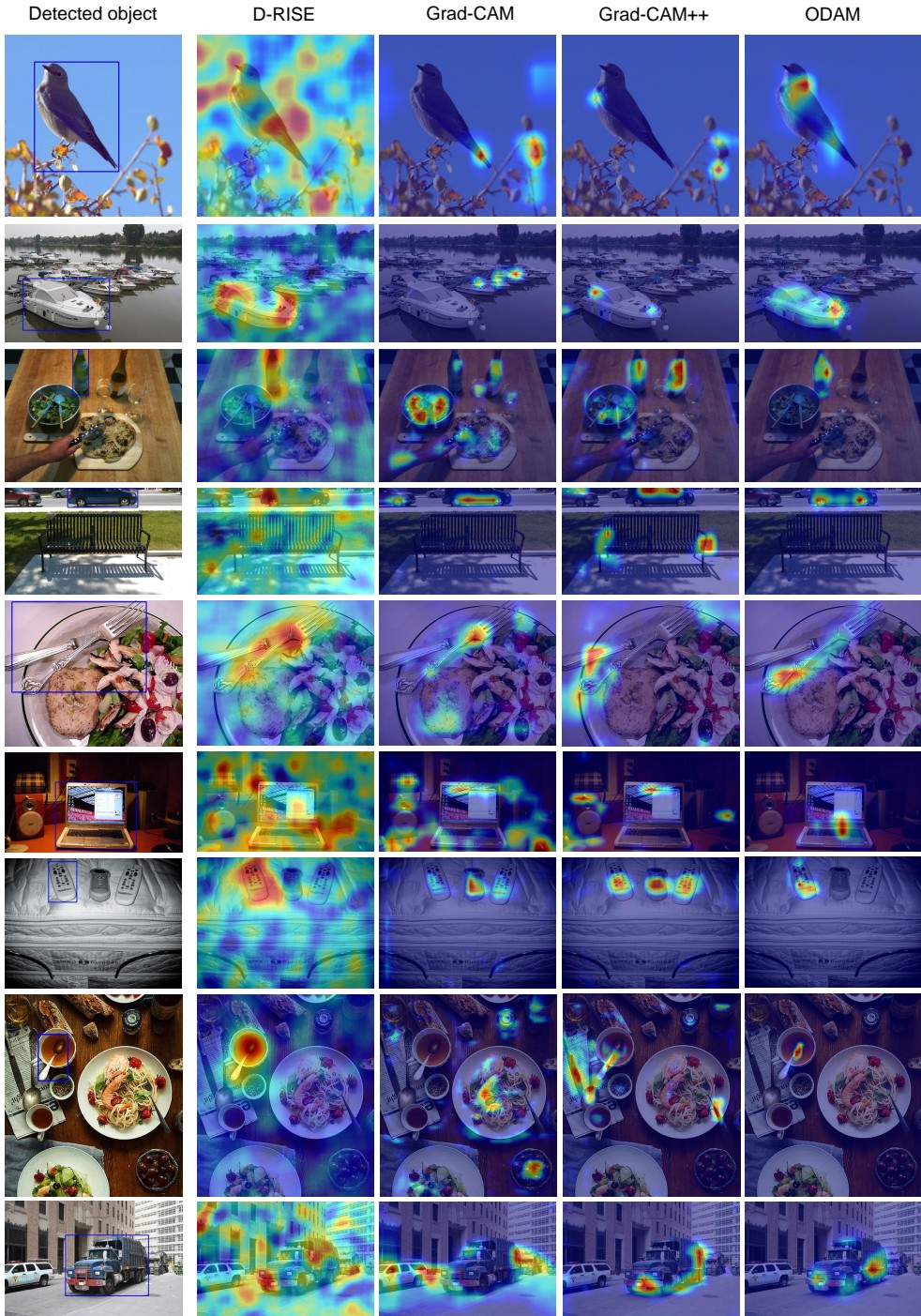

Figure 14: Visual comparison of our approach with other approaches on explaining object detection. The blue bounding box shows the explained detected object.

### A.5.5 MORE VISUALIZATIONS OF EACH DETECTION ATTRIBUTE

We provide more visualized explanations by ODAM for some high-confidence true positive predictions of the baseline FCOS detector in Fig. 15. The heat maps for the class scores and four regression values are displayed in each column.

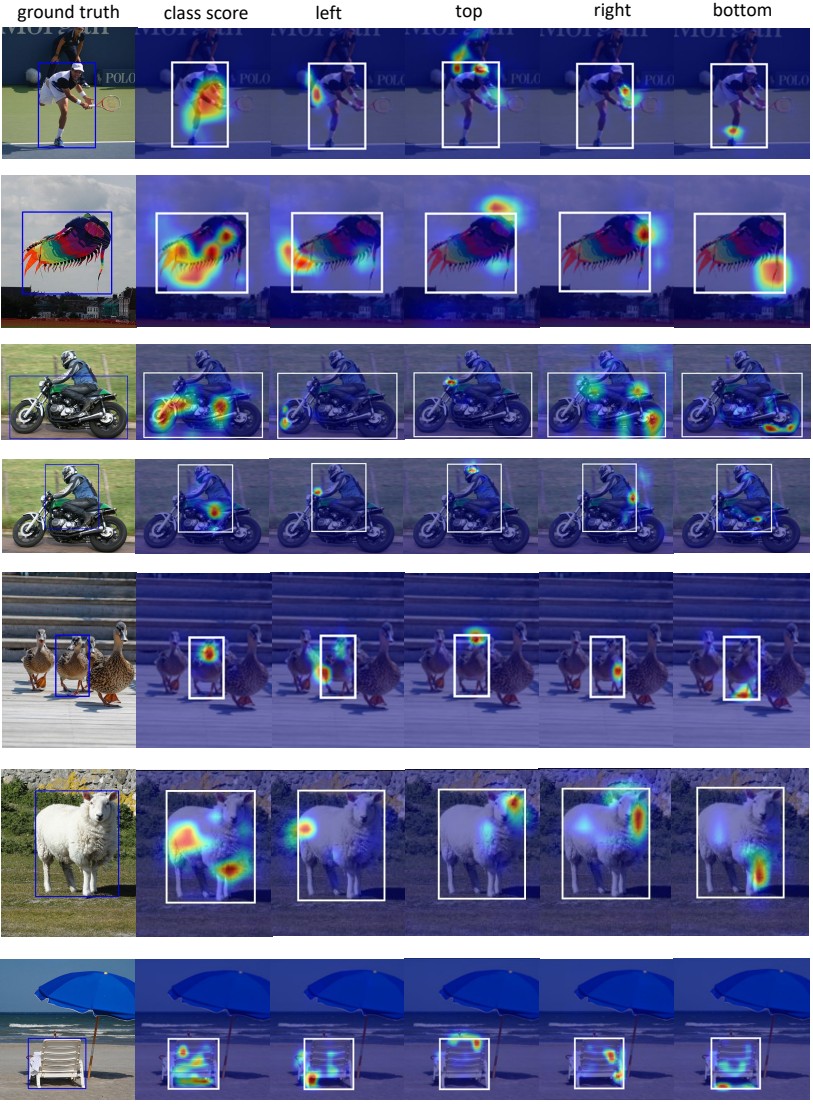

Figure 15: Visualized explanations of the class score and the regression of left($x_1$), top($y_1$), right($x_2$), and bottom($y_2$) extents, respectively.

A.5.6 THE EFFECT OF GRADIENT DIRECTION IN EXPLAINING BOUNDING BOX REGRESSION

The visualization of the bounding box regression in Fig. 3 (left) is for the FCOS detector. In FCOS, the $(x1, y1, x2, y2)$ values are the box offsets relative to the anchor center point, i.e, positive values indicate expanding the box, and negative values indicate shrinking the box. The corresponding visualizations in Fig. 3 use the ReLU to truncate the negative heat map (Eq. 2), and thus it shows features important for expanding the regressed box. These features tend to be object parts in the extremes of the object.

We can also create visualizations using other directions of the gradient or using other functions beside ReLU when computing the heat map $H$ in (2). Figure 16 shows visualizations for different settings:

a) using positive gradients and ReLU shows the features important for expanding the box, which tend to be on the extremes of the object.
b) using negative gradients and ReLU shows the feature for shrinking the box, which tend to be further inside the object.
c) using positive gradients without ReLU shows a summary of important features in either direction, using different colors.
d) using positive grardients and Abs instead of ReLU shows a summary of the most important features overall for the bounding box edge.

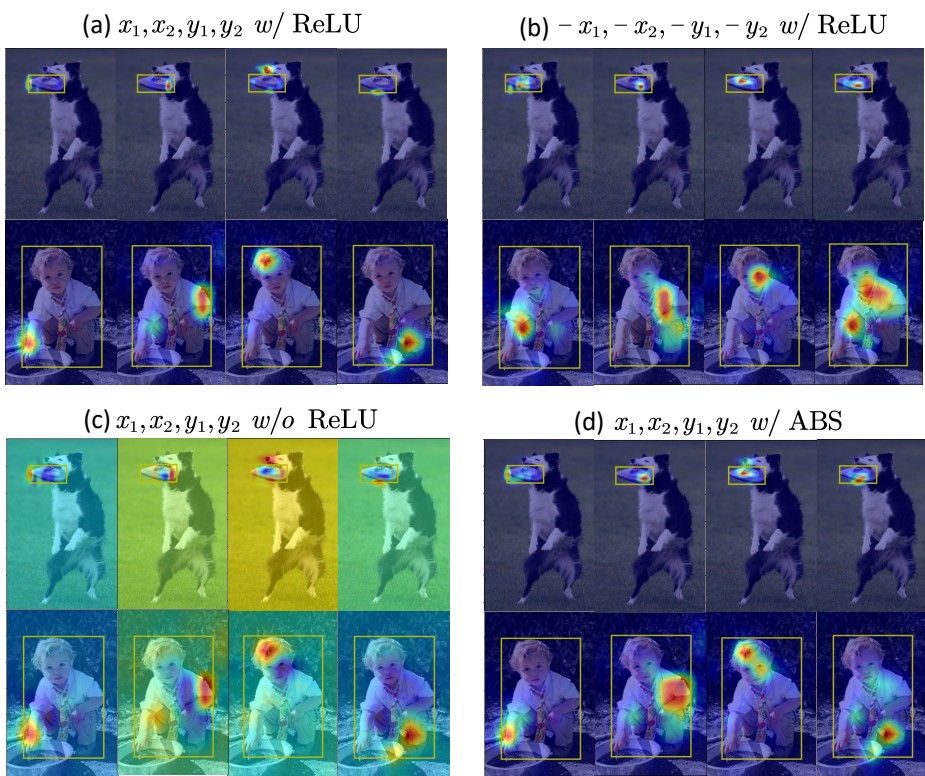

Figure 16: Heat maps of the bounding box regression $(x_1, y_1, x_2, y_2)$ under different implementation conditions.

We have conducted a user study to find out which version is most useful for users to understand how the detector has predicted the bounding box. In the questionnaire, for the predicted bounding box of an object, we show users these four kinds of explanation maps generated by ODAM. Users are asked to rank these four kinds of maps from the best explanation ($1^{st}$) to the worst explanation ($4^{th}$) with respect to the predicted bounding box. The questionnaire is composed of 20 questions with five for each box attribute (left, top, right, bottom), and was taken by 10 users, resulting in a total of 200 responses.

The percentage of ranking and average rank for each type of explanation map is shown in Tab. 10. The results indicate that the explanation map with only positive gradients better shows the user how the model predicted the bounding box, while the map with only negative gradients achieved the worst average rank (Wilcoxon signed-rank test, $p < 0.001$). The average rank of Pos w/ ReLU is significantly better than the next best method, Pos w/o ReLU (Wilcoxon signed-rank test, $p = .004$). These results confirm that the version using positive gradients for the bounding box explanation map are the most useful for users.

Table 10: User study for explaining bounding box regression using different gradient information: the percentage of ranking and average rank.

| Map type | $1^{st}$ | $2^{nd}$ | $3^{rd}$ | $4^{th}$ | Averaged Rank |
|---|---|---|---|---|---|
| (a) Pos w/ ReLU | **37.5%** | 32.0% | 17.5% | 13.0% | **2.06** |
| (b) Neg w/ ReLU | 11.5% | 17.0% | 30.5% | 41.0% | 3.01 |
| (c) Pos w/o ReLU | 32.0% | 19.0% | 27.5% | 21.5% | 2.39 |
| (d) Pos w/ Abs | 19.0% | 32.0% | 24.5% | 24.5% | 2.55 |

### A.5.7 MORE EXAMPLES FOR EXPLANATIONS FOR VARIOUS DETECTORS

Fig. 17 shows more visualizations of explanation heat maps generated with different detector heads and backbones. Interestingly, transformer-based DETR appears to be more heavily focused on object parts in the extremities of the object, while ignoring other regions, compared to the CNN-based detectors. This is likely due to the patch-set representation and transformer used as the detector head.

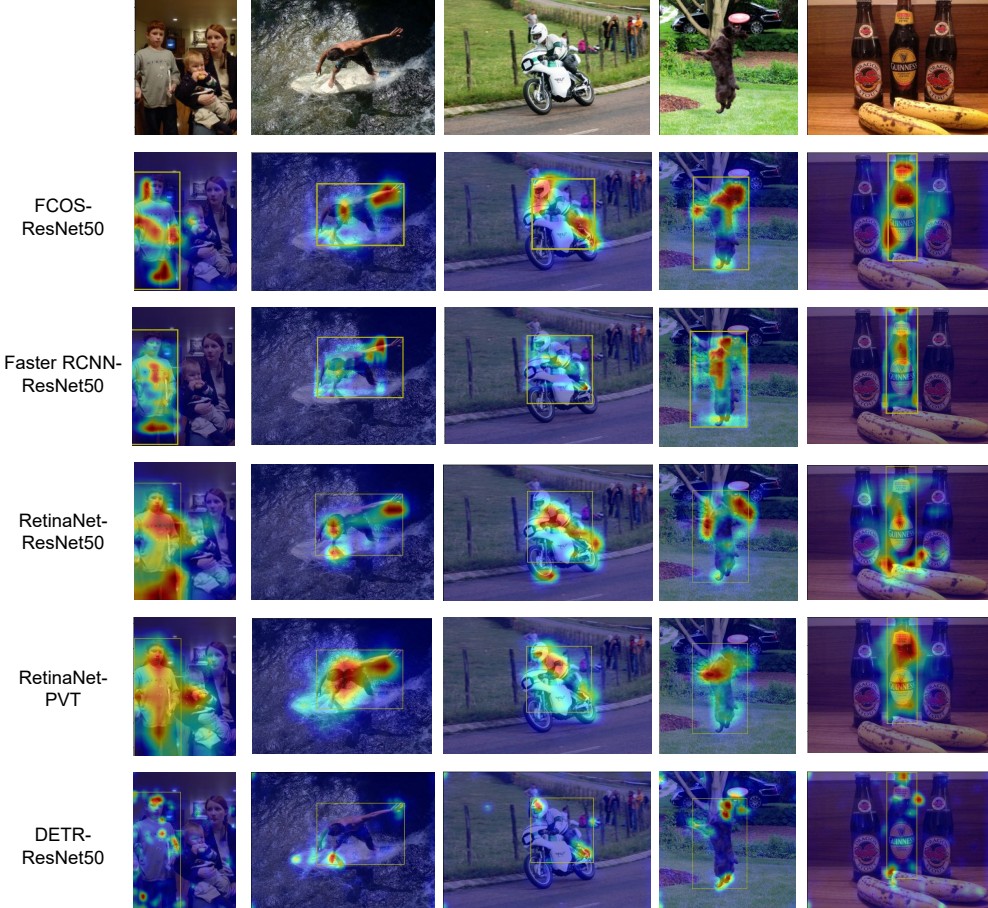

Figure 17: More explanation heat maps for various detector architectures. The combination heat maps of predicted class and bbox regression are visualized.

A.5.8   COMPARISON WITH ATTENTION MAPS IN DETR

In Figs. 18 and 19, we visualize the heat map explanations for DETR using ODAM, which is a top-down visual explanation, and the DETR transformer's self-attention, which is a bottom-up saliency map. For the encoder transformer, the self-attention will sometimes focus mainly on the object (e.g., 3rd and 4th columns of Fig. 18b), but also sometimes look at context features (e.g., in 1st and 2nd columns of Fig. 18b, the bed surrounding the cat and the remote control). For the decoder transformer shown in Fig. 18d, the self-attention will look at the extremities of the object, i.e, the points along the predicted bounding box.

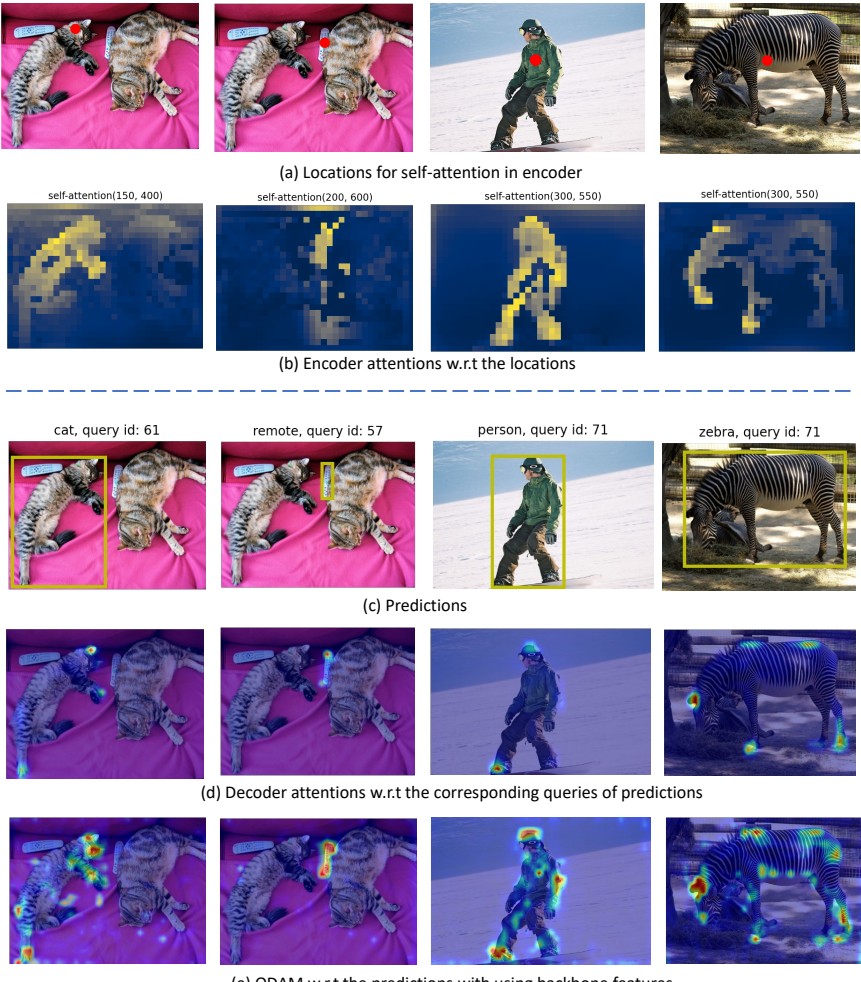

Figure 18: Visualizations of self-attention maps in DETR and heat map explanations using ODAM. (a) the locations for querying the self-attention in the encoder; (b) the encoder self-attention weights at the positions specified; (c) the predictions of the detector; (d) the decoder self-attention weights of the predictions; (e) the ODAM combo heat maps of the predictions using the backbone features.

The ODAM combo heat maps are shown in Fig. 18e, and the separate heat maps for individual attributes are shown in Fig. 19. In Fig. 19, the ODAM heat map for the *class score* is mostly consistent with the encoder self-attention maps. However in some cases (e.g., the remote control in the 2nd column), ODAM shows that less context information is actually used compared to what is indicated by the encoder self-attention map. ODAM is also able to highlight the the important regions for predicting *each coordinate* of the bounding box, e.g., the right parts of the zebra that are important for predicting the right box coordinate. In contrast, the decoder self-attention highlights all extremities of the zebra, so the self-attention itself cannot distinguish the importance for individual outputs, like the bottom coordinate of the bbox.

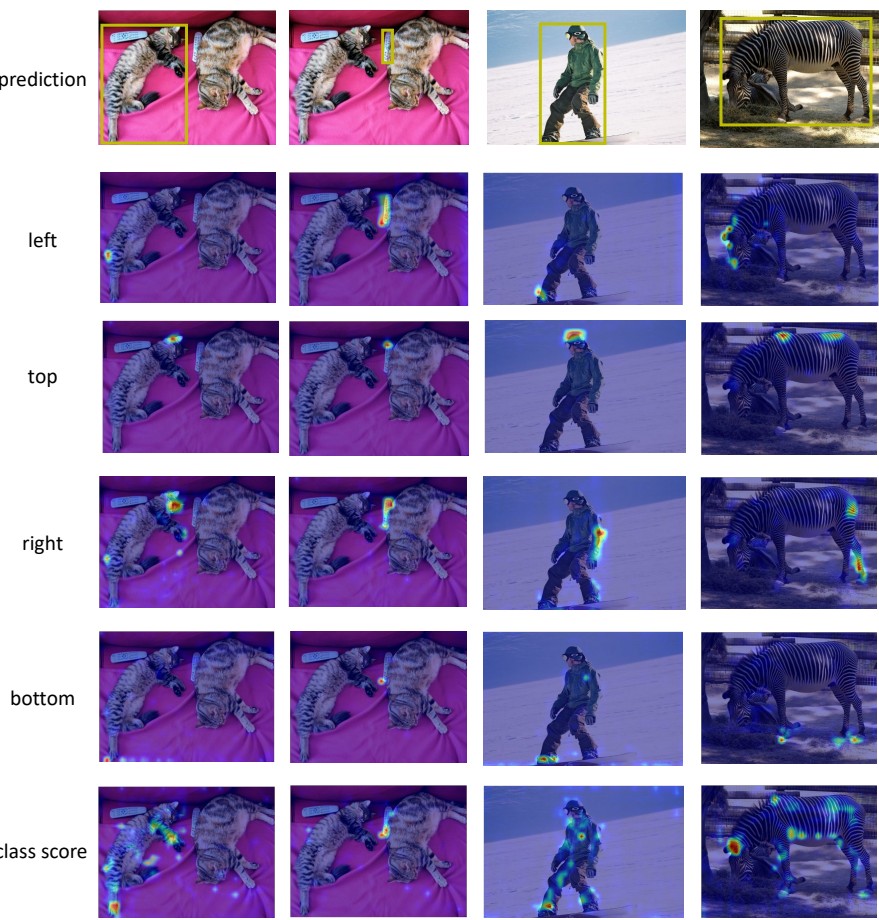

Figure 19: Visualizing the ODAM explanation heat maps for each prediction attribute of DETR, separately.

From the visualizations, we obtain some interesting interpretations of DETR: 1) the *encoder* self-attention mainly highlights the information related to object and context, but not all the context regions highlighted by self-attention are actually used for prediction according to the ODAM class heat map; 2) the *decoder* self-attention mainly highlights all regions at the extremities of the object (i.e., near the bounding box), but cannot distinguish which regions are important for a particular box coordinate. In contrast, using ODAM explanations can visualize which features are particularly important for each predicted attribute (class and each bounding box coordinate).

The transformer self-attention map is a bottom-up attention map (i.e., generally which features are interesting and correlated with the query). For example, in the transformer encoder, with the feature itself as query, heat maps highlight the attention w.r.t. each location on the feature map. In the decoder, with the query corresponding to each prediction, the attention map shows the regions that are highly correlated with the query. In contrast, the ODAM generates a top-down attention map (i.e, which features are important for the output prediction). It should be noted that for bottom-up attention, even if a feature is highlighted in the attention map, there is no guarantee that the feature is actually used in the subsequent output prediction. In contrast, for the top-down attention, all highlighted features should be important for generating the prediction. Therefore, we think that top-down visual explanations are still necessary for transformer-based detectors (Jain & Wallace, 2019), e.g., to interpret the self-attention ifself, and exploring the relationships between top-down explanations and the self-attention is an interesting area of future work.

Table 11: Comparison with BBAM using the evaluation of Deletion, Insertion, Pointing Game (PG) accuracy with ground-truth bounding boxes (b), energy-based PG with box, and Heat Map Compactness.

| | Deletion↓ | Insertion↑ | PG(b)↑ | energyPG(b) ↑ | Compactness↓ |
|---|---|---|---|---|---|
| Grad-CAM | 60.4 | 11.4 | 63.1 | 63.2 | 0.33 |
| Grad-CAM++ | 53.9 | 22.4 | 62.2 | 63.3 | 0.25 |
| BBAM | 27.5 | 48.7 | 97.0 | 96.9 | 0.059 |
| ODAM | **26.0** | **58.0** | **98.8** | **98.9** | **0.043** |

### A.5.9 COMPARISON WITH BBAM ON PASCAL VOC USING FASTER RCNN

We implemented our ODAM based on the source code of BBAM Lee et al. (2021) and generated explanation heat maps with the Faster RCNN detector framework for the max score prediction on each image using BBAM, ODAM, Grad-CAM, or Grad-CAM++ on PASCAL VOC (Everingham et al., 2010). For evaluation, we conducted the Deletion, Insertion, Pointing Game, and Compactness metrics in the same way as in Sec. 4.2

The results are shown in Tab. 11. Compared with BBAM, our ODAM performs better in terms of both faithfulness (Deletion/Insertion) and localization (PointGame/Compactness). Moreover, ODAM is over 350 times faster than BBAM; the average processing time (including model inference and heat map generation) of BBAM and ODAM for one prediction on GTX 1080Ti is 81.32s vs. 0.21s.

Qualitatively, compared to ODAM, the BBAM heat map visualizations are sparser and more focused on a minimal set of keypoints, which is due to the L1 loss of the mask. For example in Fig. 20, while ODAM highlights all the distinctive features of the head of a horse (mouth, nose, jawline, ears, eyes, forehead), BBAM will highlight only the ears and mouth.

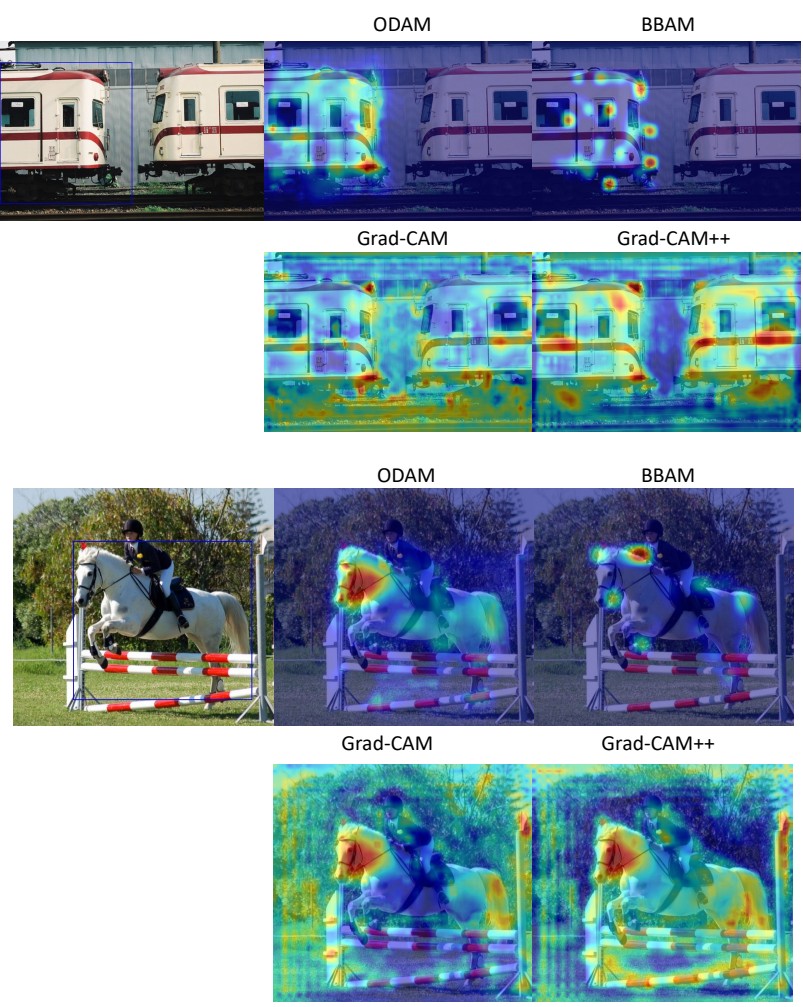

Figure 20: Visualization comparison with heatmaps from BBAM on PASCAL VOC using Faster RCNN.

