# OpenReview forum: "ODAM: Gradient-based Instance-Specific Visual Explanations for Object Detection"
_ICLR.cc/2023/Conference — ICLR 2023 poster_

### Official Review · Reviewer_k7eB · 2022-10-24

**Confidence:** 4
**Correctness:** 3
**Technical Novelty And Significance:** 3
**Empirical Novelty And Significance:** 3
**Recommendation:** 6

**Clarity, Quality, Novelty And Reproducibility:**

The ODAM is supposed to be easy to reproduce. The novelty is minor as it mainly follow the ideas of Grad-CAM.

**Strength And Weaknesses:**

Strength
- The proposed ODAM sounds technically correct to provide instance-specific heat maps for explaining the prediction attributes of object detectors, which is beyond the existing class-wise activation maps.
- The paper is easy to follow.
- OdamTrain and Odam-NMS provide insight into how to use the visual explanation to boost detection performance.

Weakness
- My main concern is that a big advantage of class-wise attention is to achieve weakly-supervised object detection or segmentation, where only image-level annotation is required. However, ODAM needs an extra bounding box to generate instance-specific maps. So can ODAM work well on a task with weak supervision? Additionally, as the authors claim that ODAM could interpret any type of prediction, I am curious how well the ODAM can perform for segmentation tasks.
- To evaluate the visual explanation quality, though the experiments look solid and extensive, I still suggest evaluating robustness, e.g., how does ODAM perform under adversarial attack?
- As the ODAM includes extra bounding box information, it seems unfair to compare ODAM with existing class-wise activation, e.g, Grad-CAM.
 - Additionally, I suggest authors include an ablation study to discuss the performance variance when ODAM is applied to different feature layers.


**Summary Of The Paper:**

The authors propose an Object Detector Activation Map (ODAM) to visually interpret object detectors' predictions. ODAM follows the ideas from Grad-CAM, using gradients to produce instance-specific heat maps highlighting the important pixels. ODAM can not only work with class labels but also bounding box coordinates. Additionally, a training scheme OdamTrain and an NMS mechanism, Odam-NMS are proposed to improve object detection performance.

**Summary Of The Review:**

To my knowledge, it is the first work to give a visual explanation of object detection. Though robustness evaluation is not included, an evaluation of visual explanation is still solid. The  ODAMTrain and ODAM-NMS achieve performance marginally boosting, providing insight into how to utilize the visual explanation. Overall, I think the paper reaches the weak acceptance bar.

---

> ### Author Response · Authors · 2022-11-16
> **Response to Reviewer k7eB:**
>
> Thanks for your insightful review. We have answered your concerns/questions here, and are happy to discuss them further. We have also updated the paper accordingly.
>
> **Q1a:** *My main concern is that a big advantage of class-wise attention is to achieve weakly-supervised object detection or segmentation, where only image-level annotation is required. However, ODAM needs an extra bounding box to generate instance-specific maps. So can ODAM work well on a task with weak supervision?*
>
> **A1a:** Sorry for the confusion. The GT bounding box information was only required for the Hiou explanation, which is used for the Hcomb map. The Hcomb map is used as a visualization to show a holistic view of the explanations between different detectors. *To be clear, we do not need the GT information (either class or bounding box) when computing the explanations for the class probability or regression coordinates.* Furthermore, in all of the quantitative evaluations, ODAM does **not** use any GT information in its calculation of explanations.
>
> We agree that using the GT box for Hiou and Hcomb is confusing and unnecessary, so in the revised paper we have changed the definition of Hcomb to combine the heat maps of individual outputs: Hcomb = max(H_class, H_x1, H_x2, H_y1, H_y2), which does **not** use any GT. The new definition appears in Sec 4.1. We have updated the visualizations in Fig. 3, and the conclusions from this visualization are the same.
>
> Regarding weakly supervised segmentation, it is an interesting direction of future work. ODAM could be used to generate heat maps inside the bounding box, which could be used as mask pseudo-labels for weakly supervised semantic and instance segmentation.
>
> **Q1b:** *Additionally, as the authors claim that ODAM could interpret any type of prediction, I am curious how well the ODAM can perform for segmentation tasks.*
>
> **A1b:** Thanks for the interesting question. We can treat the segmentation head of the detector (e.g., in Mask R-CNN) as a dense set of binary classifiers, and thus apply ODAM to each predicted segment pixel. This can explain each output segment pixel, and the explanations for all segment pixels can be aggregated to form a segment explanation. We are working on implementing this visualization and will add it later.
>
> **Q2:** *To evaluate the visual explanation quality, though the experiments look solid and extensive, I still suggest evaluating robustness, e.g., how does ODAM perform under adversarial attack?*
>
> **A2:** This is an interesting question, and we will try this for future work – using visual explanations as a method for detecting adversarial attacks on detectors.
>
> **Q3:** *As the ODAM includes extra bounding box information, it seems unfair to compare ODAM with existing class-wise activation, e.g, Grad-CAM.*
>
> **A3:** The ODAM visual explanations for the class probability (Hclass) are used in all the quantitative evaluations and user studies, and Hclass does not use the GT bounding box information. This is the same setting as Grad-CAM, and thus the comparison is fair. We have redefined Hcomb to not use the ground-truth bounding box in Hiou – see Answer A1a above. We have also made the targets for each explanation map clearer in the paper.
>
> **Q4:** *Additionally, I suggest authors include an ablation study to discuss the performance variance when ODAM is applied to different feature layers.*
>
> **A4:** Due to the page limitation, we have put the visualization of ODAM on different feature layers in Appendix A.5.3. It is a great suggestion to do a further ablation study looking at performance variations between layers -- actually our future work is to investigate which layers are best for explainability and user trust among different detectors, datasets, and explanation tasks.

---

### Official Review · Reviewer_y3xs · 2022-10-24

**Confidence:** 3
**Correctness:** 3
**Technical Novelty And Significance:** 3
**Empirical Novelty And Significance:** 3
**Recommendation:** 6

**Clarity, Quality, Novelty And Reproducibility:**

- The description of Odam-NMS needs improvement. While the intuition is clear (also with Figure 2(c)), the text only describes the algorithm vaguely. Are there multiple thresholds for the two metrics (IoU and correlation of heatmaps)? Are the two metrics combined and then thresholded with a single value? Besides pointing to the appendix for pseudo code, this information is valuable in the main paper too.
- Figure 2 is only referred to in the text in Section 3.3 for some visual examples. It should be referred to more often throughout Section 3 to help understand the heatmaps, the training losses and the NMS.

**Strength And Weaknesses:**

Strengths:
- Explainability is an important topic by itself. Methods tailed for object detection are certainly also in high demand given the wide applicability of object detection.
- The efficacy of the proposed gradient-based method over the existing perturbation-based D-RISE is good and important from a practical point of view.
- Novel non-maxima-suppression (NMS) methods that are more intelligent than looking at bounding box overlaps are certainly interesting. (Although more recent Transformer-based detectors like [A] do not require NMS anymore.) The intuition of leveraging pixels that contribute to the prediction for NMS is good; Figure 2(c) shows that well.
- Two different types of detection architectures are evaluated (one-stage, FCOS, and two-stage, Faster-RCNN)
- The quantitative evaluations in Section 4.2 are good and thorough I think.

Weaknesses:
- General comment/question regarding visual explanations and explainability
  - Shouldn't the analysis of such methods focus a lot more on erroneous predictions (both false positives and false negatives)? I assume a big part of explainability is to figure out why a model makes certain errors.
  - The qualitative inspection of heatmaps seems arbitrary and disconnected from actual "explainability". It seems the paper hints at more compact heatmaps being better at explaining the network, which feels a lot like confirmation bias. There may be regions away from the detected object that made the network predict it (e.g., contextual information).

- The motivation for Odam-Train and the Object Discrimination Index seem flawed
  - As mentioned above, the paper seems to hint at compactness of heatmaps being an indicator for a good visual explainability tool. But I do not quite understand why these two things are correlated. What if the existence of one instance of category A makes it more likely that another instance of the same category exists in the image? Another example: If the image captures the sun, wouldn't that influence the probability (confidence scores) of predicting the category "umbrella"?
  - Odam-Train essentially has the goal to make the heatmaps more compact. The loss functions make sense for that goal. But I do not understand how this makes ODAM a better explainability tool? Obviously, if we consider compactness of heatmaps as good for explaining a model, then optimizing for that will make it better. But does this really improve explainability of the model?
  - The same argument is true for the proposed Object Discrimination Index, as a quantitative measure for how good the visual explanation is. I'm wondering whether the motivation for this metric is valid.

- Odam-NMS:
  - The related paper [B] is not discussed or compared with.
  - I'm missing a quantitative comparison of Odam-NMS without using Odam-Train first. Based on the motivation of the paper, Odam-NMS without Odam-Train should perform worse.

- Qualitative results are insufficient:
  - Unfortunately, I believe the appendix or supplemental material is missing, which would include more qualitative results. I cannot find it, though. So the only qualitative results for ODAM are given in figures 1 and 3. Irrespective of the missing appendix, I would expect more visual examples in the main paper when the topic is about visual explanations.
  - As mentioned above, I think it would be important to showcase visual explanations for wrong detections.
  - The qualitative results in Section 4.1 for models trained with and without Odam-Train (Figure 4) are only a confirmation that the proposed loss is doing what it is designed for. The claim of better explainability is somewhat flawed because the loss just optimizes for the definition of the heatmap-based visual inspection tool.

- The paper mentions that ODAM is applicable to one-stage detectors like YOLO or FCOS. I am missing a discussion about recent Transformer-based (DETR-based [C]) methods like DINO [A]?

References:
- [A] DINO: DETR with Improved DeNoising Anchor Boxes for End-to-End Object Detection. Zhang et al. arXiv 2203.03605
- [B] Learning non-maximum suppression. Hosang et al. CVPR 2017
- [C] End-to-end object detection with Transformers. Carion et al. ECCV 2020

**Summary Of The Paper:**

This submission presents a framework for visual explanations of object detection models. The visual explanations are in the form of heatmaps that indicate the importance of each pixels for the prediction of an individual instance (classification score and bounding box location). Key differences to prior works are that the proposed method is instance-based (not category-based) and the heatmaps are computed gradient-based (not perturbation-based). The former makes the visual explanations more suitable for object detection, while the latter make the computation efficient. The technical contribution over gradient-based image-level visual explanation methods is that the "importance-weights" are computed for each individual pixel and activation (feature) map, while image-level methods only computed one weight per activation map. Based on the per-instance heatmaps, the paper proposes a training objective to make these heatmaps more compact, in a sense that the contributing pixels for each prediction do not overlap with other instances in the same image. Such heatmaps are then further used to improve non-maxima-suppression of standard object detectors by not only looking a intersection-over-union (IoU) between bounding boxes, but also at the correlation between the per-instance "visual explanation" heatmaps. Experiments are conducted on the COCO and CrowdHuman datasets.

**Summary Of The Review:**

Overall, I'm missing justification and for several aspects of the paper (see "Weaknesses"); might be also caused by the missing appendix.

*Update 11/28*: After author response and availability of the appendix, many of my concerns have been addressed -- raising my score.

---

> ### Author Response · Authors · 2022-11-16
> **Response to Reviewer y3xs -- part1:**
>
> Thanks for your insightful review. Sorry that we have forgotten to upload the Appendix as we had mistakenly thought it was a separate upload and deadline. The Appendix is now uploaded and has some of the answers to your questions. We have answered your concerns/questions here, and are happy to discuss it further. We have also updated the paper accordingly.
>
> **Q1a:** *General comment/question regarding visual explanations and explainability. Shouldn't the analysis of such methods focus a lot more on erroneous predictions (both false positives and false negatives)? I assume a big part of explainability is to figure out why a model makes certain errors.*
>
> **A1a:** Yes, we have put the exploration of error modes of the detector into the Appendix. Please see Appendix A.5.2, and examples in Figs. 11 and 12.
>
> **Q1b:** *The qualitative inspection of heatmaps seems arbitrary and disconnected from actual "explainability". It seems the paper hints at more compact heatmaps being better at explaining the network, which feels a lot like confirmation bias. There may be regions away from the detected object that made the network predict it (e.g., contextual information).*
>
> **A1b:** Here we are referring to explaining, *which object has been detected (i.e., how to discriminate between multiple objects)*, which is different from the standard explanation on *why the object’s class is predicted*.  For explaining object discrimination, intuitively, we want the heat map to be concentrated on the actual object, and not spread out over other objects. In the paper, we called this “compact”, but we think the term is confusing, since we don’t necessarily need the explanation to be as compact as possible for object discrimination, as long as the heat map is localized on the object itself. In the revised paper, we have changed “compactness” to “well-localized” to avoid confusion. We have quantitatively evaluated object discrimination using ODI (see Table 3), and we have also conducted a new object discrimination user study (see summary in Table 3b, and details in Table 6). Please also see Answers to Q2a and Q2b below, as well as the separate thread to all reviewers on discussion about object discrimination.
>
> In terms of our heatmap quality, we have conducted a user trust study on the original ODAM heat maps (w/o Odam-Train) in Table 2, which shows that our heat map has higher user trust. More qualitative visualizations are shown in the Appendix with A5.4 Fig.14, and A5.5 Fig.15.

---

> ### Author Response · Authors · 2022-11-16
> **Response to Reviewer y3xs -- part2:**
>
> **Q2a:** *The motivation for Odam-Train and the Object Discrimination Index seem flawed. As mentioned above, the paper seems to hint at compactness of heatmaps being an indicator for a good visual explainability tool. But I do not quite understand why these two things are correlated. What if the existence of one instance of category A makes it more likely that another instance of the same category exists in the image? Another example: If the image captures the sun, wouldn't that influence the probability (confidence scores) of predicting the category "umbrella"?*
>
> A2a: As mentioned in Answer A1b, there are two types of explanation tasks for object detectors: 1) the traditional explanation task of “what context/features are important for the class prediction?”, which we can denote as “*object specification*”; 2) the detector-specific explanation of “which object was actually detected?”, which we denote as “*object discrimination*”. For object discrimination, intuitively we would like the heat map to be well-localized on the target object, and have negligible values on the other objects. In this way, the user is certain that the correct object was localized. We hinted this as “compactness” in the paper, but the term “well-localized” is better, and we have changed it in the revised paper.
>
> Regarding context information, it is true that a detector may use existence of other objects to help detect a target object. As implied by the reviewer, the use of context information is in conflict with the ideal explanation for object discrimination, where the heat map is only over the object.  Actually, we do not think this is necessarily a problem, since they involve two different explanation tasks: (1) the context information is concerning object specification, i.e., which features were used; (2) the localization of features is concerning object discrimination, i.e., which object was detected. A detector that heavily uses context information (neighboring objects), as displayed in the heat map, will have low object discrimination ability since users cannot be sure the correct object was detected in the heat map. Furthermore, we can consider two detectors with equal detection accuracy, but one has low ODI and the other high ODI. The high ODI detector could be considered better because it does not need to use context information to perform detection, which means that it can generalize better to various scenes and is less susceptible to spurious correlations in the training dataset (e.g., objects always co-occuring during training). We have added a discussion about this in Section 5.
>
> **Q2b:** *Odam-Train essentially has the goal to make the heatmaps more compact. The loss functions make sense for that goal. But I do not understand how this makes ODAM a better explainability tool? Obviously, if we consider compactness of heatmaps as good for explaining a model, then optimizing for that will make it better. But does this really improve explainability of the model?*
>
> **A2b:** Odam-Train aims to improve the explainability of object discrimination, through encouraging the heat maps of different detected objects to be non-overlapping, and heat maps of detections of the same object to be consistent. Odam-Train does not directly make the heat maps more compact, but this is a secondary effect of making the heat maps non-overlapping. Table 3a shows that using Odam-Train improves the object discrimination (quantitatively measured via ODI), compared to without it. We have also conducted a new user study on object discrimination to show the effect of Odam-Train on object discrimination. Given a heat map and image, the user is asked to draw a box around the object they think the detector found, and also state their confidence. The results are presented in Table 3b in Section 4.2, and Table 6, Figure 8 of Appendix A.3. Using Odam-Train with ODAM significantly increases the user’s confidence (from 71% level-5 confidence to 84%), as compared to without using Odam-Train. The accuracy of the user’s box also increases slightly when using Odam-Train. Comparing with the other methods, ODAM has higher confidence and accuracy than previous works, which is consistent with the ODI results in Table 3a. In summary, the user study shows that ODAM has better object discrimination than other methods, and using Odam-Train can further improve the user’s confidence in the detector’s explanation map for discriminating the object. We have updated the paper and Appendix with the new user study on object discrimination.
>
> Finally, we note that using Odam-Train has negligible effect on the detector accuracy, as shown in Tables 8 and 9 in Appendix A4.3.

---

> ### Author Response · Authors · 2022-11-16
> **Response to Reviewer y3xs -- part3:**
>
> **Q2c:** *The same argument is true for the proposed Object Discrimination Index, as a quantitative measure for how good the visual explanation is. I'm wondering whether the motivation for this metric is valid.*
>
> **A2c:** ODI measures how good the visual explanation is for *object discrimination*, which is different from explaining the features for classification (object specification). For low ODI, the detector is able to explain its detections using features that are mainly on the actual object, and not based on other object features. Thus, users will be able to better discriminate which object was detected. We confirm this in a new user study on object discrimination (Table 3b, Table 6), where our method with low ODI achieves high confidence and accuracy from the user in an object discrimination task (Table 3b).
>
> **Q3a:** *Odam-NMS: The related paper [B] is not discussed or compared with. [B] “Learning non-maximum suppression.} Hosang et al. CVPR 2017*
>
> **A3a:** Thanks for the reference. [B] introduces an extra convolutional network Gnet after the detector to rescore the predictions, and trains it to decrease the score of predictions that may heavily overlap with others. Unfortunately, [B] did not provide results on crowded datasets, such as CrowdHuman (they report results on PETS and COCO person miniset), nor did they provide a concrete architecture for the Gnet or code for the training details. Therefore, it is hard to compare with [B] in a short time. We have included this reference in Section 2, and will try to implement it later.
>
> **Q3b:** *I'm missing a quantitative comparison of Odam-NMS without using Odam-Train first. Based on the motivation of the paper, Odam-NMS without Odam-Train should perform worse.*
>
> **A3b:** The accuracy comparison of Odam-NMS with and without Odam-Train is in Appendix A.4.3, Tables 8 and 9. Using Odam-NMS with Odam-Train improves the JI and MR compared to without Odam-Train (Table 9). Odam-Train also improves the recall on crowded scenes (Table 8).
>
> **Q4a:** *Qualitative results are insufficient. Unfortunately, I believe the appendix or supplemental material is missing, which would include more qualitative results. I cannot find it, though. So the only qualitative results for ODAM are given in figures 1 and 3. Irrespective of the missing appendix, I would expect more visual examples in the main paper when the topic is about visual explanations.*
>
> **A4a:** Additional visualizations are shown in the Appendix A.5, specifically Figures 11-17. We will add more examples to the main paper if space allows.
>
> **Q4b:** *As mentioned above, I think it would be important to showcase visual explanations for wrong detections.*
>
> **A4b:** Please see Appendix A.5.2, Figures 11 and 12, for examples of wrong detections and the insight revealed from them.
>
> **Q4c:** *The qualitative results in Section 4.1 for models trained with and without Odam-Train (Figure 4) are only a confirmation that the proposed loss is doing what it is designed for. The claim of better explainability is somewhat flawed because the loss just optimizes for the definition of the heatmap-based visual inspection tool.*
>
> **A4c:** We agree that the Odam-Train objective of minimizing overlap of heatmaps from different objects is aligned with the ODI metric, which computes the proportion of heatmap energy within other object boxes. To show that ODI is a valid metric for user-centric explainability, we have conducted a new user study on objection discrimination in Appendix A.3. Given a heat map and image, the user is asked to draw a box around the object they think the detector found, and also state their confidence. The results are presented in Table 6 in Appendix A.3. Using Odam-Train with ODAM significantly increases the user’s confidence (from 71% level-5 confidence to 84%), as compared to without using Odam-Train. The accuracy of the user’s box also increases slightly when using Odam-Train.  Comparing with the other methods, ODAM has higher confidence and accuracy than previous works, which is consistent with the ODI results in Table 3a. Thus, our optimization using Odam-Train indeed improves the user confidence and explainability of object discrimination.  We have summarized these results in Sec 4.2 Object Discrimination and Table 3b.

---

> ### Author Response · Authors · 2022-11-16
> **Response to Reviewer y3xs -- part4:**
>
> **Q5:** *The paper mentions that ODAM is applicable to one-stage detectors like YOLO or FCOS. I am missing a discussion about recent Transformer-based (DETR-based [C]) methods like DINO [A]? [A] “DINO: DETR with Improved DeNoising Anchor Boxes for End-to-End Object Detection.” Zhang et al. arXiv 2203.03605; [C] “End-to-end object detection with Transformers.” Carion et al. ECCV 2020*
>
> **A5:** Please see the separate response thread to all reviewers about applying ODAM to transformer-based detectors.
>
> **Q6:** *The description of Odam-NMS needs improvement. While the intuition is clear (also with Figure 2(c)), the text only describes the algorithm vaguely. Are there multiple thresholds for the two metrics (IoU and correlation of heatmaps)? Are the two metrics combined and then thresholded with a single value? Besides pointing to the appendix for pseudo code, this information is valuable in the main paper too.*
>
> **A6:** The pseudo-code for Odam-NMS is in the Appendix A.1. There is one threshold for IoU, and a low- and high-threshold for correlation. The values are thresholded separately. We have added these details to the paper in Sec 3.3.
>
> **Q7:** *Figure 2 is only referred to in the text in Section 3.3 for some visual examples. It should be referred to more often throughout Section 3 to help understand the heatmaps, the training losses, and the NMS.*
>
> **A7:** Thanks for the suggestion. We have updated the paper as suggested.

---

> ### Comment · Reviewer_y3xs · 2022-11-28
> **Re: Response to Reviewer y3xs**
>
> Dear authors.
>
> I very much appreciate the detailed explanations, clarifications, and additional experiments. These answers along with the appendix helped me to better understand the paper, especially regarding my misunderstanding of object *discrimination* explanations. I think most of my concerns have been resolved now. Two more comments:
>
> * The additional user study in Appendix A3 is great. I would highlight this further (if not already done ... I didn't go through the whole paper)
> * Figure 12 may not be the best examples to showcase. For the first example, it seems both predictions are correct, no? The textual description only hypothesizes about a potential issue with the detector that it does not leverage the unique features of a laptop keyboard (although I'd argue that the trackpad is not part of the keyboard). In the second example, the "misclassified" prediction is just a reflection in the glass. Is that even part of the real ground truth? If no, it's not only misclassified as "sink" but even before that it should not have been detected as any object, no? Maybe there are better examples to showcase, and in any case, please add the actual ground truth in the visualization.

---

> > ### Author Response · Authors · 2022-12-02
> > **Thanks for your response! We appreciate your further suggestions.**
> >
> > 1. Yes, the summary of the new user study appears in Table 3 and Sec 4.2 Object Discrimination.
> >
> > 2. For Figure 12, we have found some better examples and will update the figure. In one example, the model correctly classifies an instance as “bed” when seeing the cushion of the bed, but also mistakenly predicts “bench” based on a long metal bed frame at the end of the bed. In another example, a person is fixing a wheel on the ground, and two motorcycles are parked nearby. The detector correctly finds the person, but also mistakenly detects a motorcycle on the person, by combining the features from the two motorcycles. This shows a failure mode of the detector, where sometimes the context feature (a person next to unrelated motorcycle parts) may bring negative influence to the detection result.

---

### Official Review · Reviewer_EVbq · 2022-10-25

**Confidence:** 4
**Correctness:** 3
**Technical Novelty And Significance:** 2
**Empirical Novelty And Significance:** 3
**Recommendation:** 6

**Clarity, Quality, Novelty And Reproducibility:**

The paper is clearly written and easy to understand. In my opinion, the proposed method can be easily reproduced. However, I think the novelty is quite limited.

**Strength And Weaknesses:**

Strength
1. The proposed method is sound and intuitive.

2. The experiments are thorough and show the effectiveness of the proposed method.

3. The generated explanation can be utilized to improve NMS

Weakness
1. My major concern is a technical novelty. The proposed method is simple and effective, but naive. The modification of Grad-CAM seems straightforward.

2. I am not sure if ODAM-Train can provide "real" explanation of the neural network. It is natural for different predictions to produce different interpretations, but ODAM-Train forces the explanations of different predictions to be consistent. Although this may improve several numerical results, it is questionable whether it can provide real explanations of neural networks.

-- In Figure 1, ODAM-Train seems focus on the center of the object, but intuitively thinking, contour of the object is more important than the center for object detection.

-- Please provide the results of deletion and insertion experiments of ODAM-Train

3. Due to the ReLU operator in Eq (2), the gradient direction is very important for Grad-CAM. Gradient direction of the classification score is quite intuitive (which feature is important to increase the classification score), but the gradient direction of bounding box regression is ambiguous. Which gradient should be considered? The gradient that increases the value of box regression? Or gradient that decreases the value of box regression? The increase of values of x1 and y1 shrinks the bounding box, and the increase of values of x2 and y2 expands the bounding box.

4. There are some missing references. [A] also aims to interpret the object detectors. [B] also obtains a visual saliency map for object detectors. In particular, [B] can be directly compared to the proposed method. Please provide the comparison with [B].

[A] Wu, Tianfu, and Xi Song. "Towards interpretable object detection by unfolding latent structures." ICCV 2019.

[B] Lee, Jungbeom, et al. "Bbam: Bounding box attribution map for weakly supervised semantic and instance segmentation." CVPR 2021.

5. Following recent trends, I recommend the authors to apply the method to transformer-based object detectors like DETR. In addition, the transformer-based methods provide attention map, which also can be considered as an interpretation map. Please provide the detailed comparison with those methods.

**Summary Of The Paper:**

This paper proposes an interpretation method for object detector. The authors slightly modify the existing Grad-CAM method to make it location-sensitive. In addition, ODAM-Train is also proposed for improved interpretation for overlapping objects. The experiments show that the proposed method provides better interpretation of object detector than previous methods, and can be applied to better NMS method.

**Summary Of The Review:**

This paper is well-written and well-motivated. However, I have major concerns as mentioned above. If the authors can address my concerns, I am open to increase my score.

---

> ### Author Response · Authors · 2022-11-16
> **Response to Reviewer EVbq -- part1:**
>
> Thanks for your insightful review. We have answered your concerns/questions here, and are happy to discuss it further. We have also updated the paper accordingly.
>
> **Q1:** *My major concern is technical novelty. The proposed method is simple and effective, but naive. The modification of Grad-CAM seems straightforward.*
>
> **A1:** As far as we know, this is the first successful attempt to extend gradient-based explanations, e.g., Grad-CAM, from only explaining image classification results to explaining object detection results. Although the proposed ODAM is straightforward and easy to implement, it is **significantly more effective and more efficient** compared with the previous work D-RISE. Specifically, ODAM is significantly better than D-RISE on **all** evaluations, including faithfulness (insertion, deletion, VEA in Table 1), object discrimination (ODI, pointing game, compactness in Tables 3a and 4), and user trust studies (explanations in Table 2, and object discrimination in Table 3b, Table 6). Furthermore, ODAM takes only 2 ms to compute one heat map, whereas D-RISE takes 3 minutes (over 90,000 times slower). Therefore, ODAM has large potential impact since it is easy to adopt, effective, and very efficient. Simplicity is not always bad, since it is a desirable feature to promote its adoption.
>
> Besides ODAM, we also propose a novel training scheme and corresponding NMS algorithm, Odam-Train and Odam-NMS, to improve the explanation for object discrimination of the detector.
>
> Our contributions are summarized at the end of Section 1.
>
> **Q2:** *I am not sure if ODAM-Train can provide "real" explanation of the neural network. It is natural for different predictions to produce different interpretations, but ODAM-Train forces the explanations of different predictions to be consistent. Although this may improve several numerical results, it is questionable whether it can provide real explanations of neural networks.*
>
> **A2:** To be clear, ODAM-train encourages the explanations of the predictions of the same GT object to be consistent. Here the consistency is based on the best prediction of the object that has highest IoU with the GT bbox. After using Odam-Train, the object discrimination ability of the explanations is better. Table 3 shows that using Odam-Train improves the quantitative metric of object discrimination (ODI), compared to without it. We also conduct a new user study on object discrimination, which shows that Odam-Train also improves user confidence to discriminate the object in the heat map from other objects. Therefore, we think that Odam-Train is indeed providing real explanations that improve user confidence in the detector. The summary is added to Table 3(b) and Sec 4.2 Object Discrimination, and the details are added to Appendix A.3 and Table 6, and Figure 8.
>
> Finally, we also note that our setting using Odam-Train is fair, since supervising the explanation maps will affect the **whole** detector pipeline (due to the calculated top-down gradients). Thus, the explanation maps that are generated after using Odam-Train are an accurate reflection of the detector’s inner workings. We have added a sentence about this at the end of Sec 3.2.
>
> For more details, please see the separate response thread to reviewers about object discrimination and Odam-Train.
>
> **Q2a:** *In Figure 1, ODAM-Train seems to focus on the center of the object, but intuitively thinking, the contour of the object is more important than the center for object detection.*
>
> **A2a:** The contour of the object is likely important for regressing the bounding box, e.g, see the explanation maps for x1, y1, x2, and y2 in Fig 3 (left). The heat maps in Figure 1 are for the object class probability. The maps without ODAM-train tend to focus on key points of the object, e.g. torso and leg of the person. For the model trained with ODAM-train, the maps do tend to focus more on the central portion of the object, which can improve the object discrimination ability. Note that features from the central part of the object can still “see” the extremities of the object due to the receptive field of the feature map. Thus, the accuracy of the detector is not affected much, as shown in Tables 8 and 9 in Appendix A.4.3.
>
> **Q2b:** *Please provide the results of deletion and insertion experiments of ODAM-Train*
>
> **A2b:** We have added the faithfulness results for Odam-Train to Table 1 and Figure 5. Using Odam-Train will make the faithfulness metrics slightly worse than ODAM w/o Odam-Train, but still better than D-RISE in Insertion and VEA. On the other hand, using Odam-Train results in better object discrimination, both in quantitative ODI and in the new user study (Tables 3 and 6). Interestingly, there seems to be a tradeoff between faithfulness and object discrimination, since the latter aims to only use pixels localized on the object. Future work could consider how to diminish this tradeoff. We have added a short discussion Sec 5.

---

> ### Author Response · Authors · 2022-11-16
> **Response to Reviewer EVbq -- part2:**
>
> **Q3:** *Due to the ReLU operator in Eq (2), the gradient direction is very important for Grad-CAM. Gradient direction of the classification score is quite intuitive (which feature is important to increase the classification score), but the gradient direction of bounding box regression is ambiguous. Which gradient should be considered? The gradient that increases the value of box regression? Or gradient that decreases the value of box regression? The increase of values of x1 and y1 shrinks the bounding box, and the increase of values of x2 and y2 expands the bounding box.*
>
> **A3:** This is a great point, and we thank the reviewer for their insight. The visualization of the bounding box regression in Fig. 3(left) is for the FCOS detector. In FCOS, the (x1,y1,x2,y2) values are the box offsets relative to the anchor center point, i.e, positive values indicate expanding the box, and negative values indicate shrinking the box. The corresponding visualizations in Fig. 3 use the ReLU to truncate the negative heat map (Eq 2), and thus it shows features important for expanding the regressed box. These features tend to be object parts in the extremes of the object.
>
> As mentioned by the reviewer, we can also create visualizations using other directions of the gradient or using other functions besides ReLU when computing the final heat map H in (2). In Appendix A.5.6 and its Fig. 16, we have added visualizations of the negative gradients of (x1,x2,y1,y2), which shows which features will shrink the bounding box; without using the ReLU in (2), which summarizes features for each direction; and using Abs instead of ReLU, which summarizes which features are overall important for the bounding box edge. Please see the details in Appendix A.5.6.

---

> ### Author Response · Authors · 2022-11-16
> **Response to Reviewer EVbq -- part3:**
>
> **Q4:** *There are some missing references. [A] also aims to interpret the object detectors. [B] also obtains a visual saliency map for object detectors. In particular, [B] can be directly compared to the proposed method. Please provide the comparison with [B].*
>
> *[A] Wu, Tianfu, and Xi Song. "Towards interpretable object detection by unfolding latent structures." ICCV 2019.*
>
> *[B] Lee, Jungbeom, et al. "Bbam: Bounding box attribution map for weakly supervised semantic and instance segmentation." CVPR 2021.*
>
> **A4:** Thank you for the additional references. We have added these references to the paper in Section 2.
>
> [A] This paper focuses on generating a latent-part model for each detected instance, i.e., an explanation comprising a high-level structure of cohesive object parts. In contrast, ODAM is a different explanation task, which is to identify important regions for making the detector prediction.
>
> [B] Thanks for introducing this missing visual saliency method. BBAM is a perturbation-based method, which defines an optimization problem to find a minimal mask on the image while trying to keep the detector prediction the same (either class probability or bounding box). An L1 loss on the mask encourages the mask to be as small as possible, while a loss on the detector output for the masked image tries to keep the prediction the same. BBAM has several disadvantages as a visual explanation method: 1) the L1 loss on the mask makes the mask very sparse and tend to take binary values (0,1), and it’s unclear if this is a suitable as a saliency visualization, which aims to rank pixels by importance; 2) searching for a minimal mask may throw away redundant information that is also used by the detector (e.g., other wheels of a car), and thus their map will not be a complete representation of the detector's strategy; 3) in practice, the BBAM optimization problem is solved using ADAM for 300 iterations, which will be very slow since the gradients need to be back-propagated to the image layer 300 times (at least 300 times slower than ODAM); 4) the BBAM optimization problem is essentially a feature selection problem on the pixel space, which likely has multiple local minima, and so the generated result will depend on the initialization and could result in noise. Therefore, it is unclear whether the produced mask will be good as a heat map explanation, and the paper’s original purpose is to generate instance pseudo-labels for weakly supervised semantic/instance segmentation. We are working on implementing BBAM,  but we may not be able to finish before the discussion deadline, since the original BBAM is based on 2-stage detectors (public implementation on Faster-RCNN and MASK-RCNN), and thus we must extend BBAM to 1-stage detectors and compare it based on the same detector and model.  In any case, we will add the BBAM results to the final paper. Also, the usage of instance heat map of detector on weakly supervised segmentation is an interesting direction of future work, and we will explore this later.
>
> **Q5:** *Following recent trends, I recommend the authors apply the method to transformer-based object detectors like DETR. In addition, the transformer-based methods provide attention map, which also can be considered as an interpretation map. Please provide the detailed comparison with those methods.*
>
> **A5:** Please see the separate response thread to all reviewers about applying ODAM to transformer-based detectors.

---

> > ### Comment · Reviewer_EVbq · 2022-11-28
> > **Response to author feedback**
> >
> > I really appreciate that the authors carefully responded to the reviewers' comments. However, I still have some concerns.
> >
> > - I am still not convinced that ODAM-Train provides real explanations of object detectors. ODI and user study favors an ability to accurately localize target object, but the ability does not guarantee that the generated saliency map actually explain the output of the object detector. For example, if we obtain localization of the target object using Grab-Cut, or any other salient object detectors, these maps may produce significantly better ODI and user study results, but it has nothing to do with the explanation of the object detector.
> > In my opinioin, deletion and insertion experiments, rather than ODI or user studies, indicate more effectively whether the generated saliency maps actually find important regions to the neural network. However, ODAM-Train actually reduces the performance of Deletion and Insertion experiments according to the updated Figure 5 and Table 1.
> >
> >
> > - I thank the authors for the analysis of gradient directions. However, it is still unclear which direction should be considered as the final explanation of object detectors.
> >
> > - I recommend the authors implement ODAM for 2-stage detector, rather than implementing BBAM for 1-stage detector, becuase this can highlight the generality of ODAM if ODAM can also be effectively applied to 2-stage detector. 2-stage detectors seem outdated, but still actively used in many fields and applications.

---

> > > ### Author Response · Authors · 2022-11-30
> > > **Response to the remaining concerns -- part1**
> > >
> > > Thanks for raising your remaining concerns. Please see our responses below. We will update the paper accordingly.
> > >
> > > **Q1a.** *I am still not convinced that ODAM-Train provides real explanations of object detectors. ODI and user study favors an ability to accurately localize target object, but the ability does not guarantee that the generated saliency map actually explain the output of the object detector. For example, if we obtain localization of the target object using Grab-Cut, or any other salient object detectors, these maps may produce significantly better ODI and user study results, but it has nothing to do with the explanation of the object detector.*
> > >
> > > **A:** We are sorry that we may have misunderstood your earlier comment “Although this may improve several numerical results, it is questionable whether it can provide real explanations of neural networks.” Here we thought “real explanation” referred to “explanations useful for real user”, which we answered by conducting the new user study on object discrimination. We agree that using Grab-Cut to generate “explanations” is not appropriate, since Grab-Cut does not use the detector internal representations, and thus it is not a real explanation of the detector. In contrast, we think that our Odam-Train is entirely appropriate, since Odam-Train is directly supervising the detector through the detector’s own explanations generated during training, and thus the generated explanations are real and well represent the detector’s internal process. We note that analogous work in VQA has also trained their model through the top-down explanation maps [A]. We provide more detailed analysis below.
> > >
> > > ODAM-Train is supervising the detector through its ODAM heat maps generated during training. Hence the generated ODAM maps are real explanations of the detector itself. More specifically, Odam-Train uses our auxiliary loss functions to apply supervision on the ODAM maps generated from the detector during training (Eq. 3). Since the ODAM maps are an explanation of the detector’s process that is calculated from the detector’s parameters, Odam-Train will train the detector such that its process’s explanation reduces the auxiliary loss. Specifically, Odam-Train will train the detector to predict the object’s class using features that are not overlapped with other objects’ features. Note that the generation of the ODAM maps from the detector has no trainable parameters (Eq. 2), so Odam-Train purely trains the detector itself, and there is no “cheating” to improve the generated explanation map outside of the detector.
> > >
> > > In more detail, denote our auxiliary loss function as $L(H)$, where $H$ is the heat map. During training, the gradient of the auxiliary loss w.r.t. the detector parameters $\theta$ is:
> > > 	$$\frac{\partial L}{\partial \theta} = \sum_{ij} \frac{\partial L}{\partial H_{ij}} \frac{\partial H_{ij}}{\partial \theta}$$
> > > where $H_{ij}$ is the heat map at location ${ij}$. We will examine the 2nd term. From (2), the heat map is
> > > $$H_{ij} = ReLU(\sum_k w_{ijk} A_{ijk}),$$
> > > where $A_{ijk}$ is the activation at location $ij$ and channel $k$, and $w_{ijk}=\frac{\partial y}{\partial A_{ijk}}$ is the corresponding pixel weight for output $y$.  Thus we obtain the gradient of the heat map as,
> > > 	$$\frac{\partial H_{ij}}{\partial \theta} = \begin{cases} 0, & H_{ij} \leq 0, \\\\
> > > 		 \sum_k \Big[w_{ijk} \frac{\partial A_{ijk}}{\partial \theta} + \frac{\partial w_{ijk}}{\partial \theta} A_{ijk}\Big], & H_{ij} > 0. \end{cases}$$
> > > Thus optimization of the Odam-Train loss through the ODAM heat map will affect the activations $A_{ijk}$ and the weight $w_{ijk}$ in the detector. Note that $w_{ijk}=\frac{\partial y}{\partial A_{ijk}}$ is the importance of the feature $A_{ijk}$ on the detector output $y$. Here the Odam-Train supervision encourages the usage of particular features and also increases the importance of those features in generating the output.
> > >
> > > [A] R. R. Selvaraju, S. Lee, Y. Shen, H. Jin, S. Ghosh, L. Heck, D. Batra, and D. Parikh, “Taking a hint: Leveraging explanations to make vision and language models more grounded,” in ICCV, 2019, pp. 2591–2600.

---

> > > ### Author Response · Authors · 2022-11-30
> > > **Response to the remaining concerns -- part2**
> > >
> > > **Q1b:** *In my opinion, deletion and insertion experiments, rather than ODI or user studies, indicate more effectively whether the generated saliency maps actually find important regions to the neural network. However, ODAM-Train actually reduces the performance of Deletion and Insertion experiments according to the updated Figure 5 and Table 1.*
> > >
> > > **A:** Generating saliency maps of important regions for the NN’s output prediction is the traditional explanation task for image classification. We have already used Deletion/Insertion/User Trust in Sec. 4.2 (Tab.1 and Tab.2) to evaluate the faithfulness of the ODAM explanations for object class predictions. However, besides evaluating on the traditional explanation task, we propose a new detection-specific explanation task, object discrimination, which is to explain “what object was actually detected?”  The bounding box of a detected object will sometimes be ambiguous when there are overlapping objects. For example, in Fig 4(a), from the bounding box itself, it is difficult to tell which person was actually detected in the first 2 boxes. The object discrimination task is thus to use the explanation heat map to explain which object was detected for each bounding box. Using Odam-Train improves the object discrimination ability of the explanation map, by training the detector to focus more on central features of the object that are not overlapped with other objects (see Fig. 4(b)).  Indeed Odam-Train slightly reduces the faithfulness performance, since the detector now focuses more on an object’s central features, but the object discrimination is improved (as measured with ODI and the user study). How to manage the tradeoff between faithfulness explanations and object discrimination explanations is an interesting area of future work. Additionally, we note that we have used Odam-Train with Odam-NMS to demonstrate the efficacy of object discrimination ability for improving NMS in crowded scenes with overlapped objects.
> > >
> > > **Q2:** *I thank the authors for the analysis of gradient directions. However, it is still unclear which direction should be considered as the final explanation of object detectors.*
> > >
> > > **A:** For explaining the bounding box regression, we think that using the positive gradients is sufficient for users, since the resulting active regions in the heat map are near the bounding box edges, showing which features can support the bounding box definition. We will conduct a user study to compare the 4 types of bbox explanations to see which is most useful for users to understand how the detector has predicted the box.
> > >
> > > **Q3:** *I recommend the authors implement ODAM for 2-stage detector, rather than implementing BBAM for 1-stage detector.*
> > >
> > > **A:** Thanks for the recommendation. We have already implemented ODAM for the 2-stage detector Faster-RCNN on MSCOCO, and the combo heat map is shown in Fig. 3 (right) and Fig. 17. The source codes of BBAM only provided a trained model of Faster RCNN on PASCAL VOC dataset that is adaptive to their implementation. For consistency with our extensive experiments on MSCOCO, we would like to run BBAM on MSCOCO too. As suggested by the reviewer, we will work on generating the ODAM results on Pascal VOC for comparing with BBAM first, and add them to the paper.

---

> > > ### Author Response · Authors · 2022-12-07
> > > **Comparison with BBAM on PASCAL VOC dataset for Faster RCNN**
> > >
> > > We implemented our ODAM based on the source code of BBAM and generated explanation heat maps for the max score prediction on each image using BBAM, ODAM, Grad-CAM, or Grad-CAM++ on PASCAL VOC. For evaluation, we conducted the Deletion, Insertion, PointGame, and Compactness metrics in the same way as in Sec. 4.2. Here are the results:
> > >
> > > |Deletion| Averaged classification score vs. deletion steps (0:10:100) | AUC $\downarrow$|
> > > |  ----  |  ----  | ----  |
> > > |BBAM| 1.0000, 0.8216, 0.6254, 0.4410, 0.2927, 0.1972, 0.1428, 0.0964, 0.0614, 0.0411, 0.0306 | 0.275 |
> > > |ODAM| 1.0000, 0.8064, 0.5989, 0.4175, 0.2832, 0.1767, 0.1124, 0.0816, 0.0560, 0.0376, 0.0260 | **0.260** |
> > > |Grad-CAM| 1.0000, 0.8856, 0.7920, 0.7159, 0.6537, 0.6003, 0.5485, 0.5060, 0.4737, 0.4465, 0.4217 | 0.604 |
> > > |Grad-CAM++| 1.0000, 0.8959, 0.7807, 0.6724, 0.5900, 0.5178, 0.4574, 0.4152, 0.3792,0.3505, 0.3290 | 0.539 |
> > >
> > > |Insertion| Averaged classification score vs. insertion steps (0:10:100) | AUC $\uparrow$|
> > > |  ----  |  ----  | ----  |
> > > |BBAM| 0.0000, 0.0000, 0.0145, 0.1641, 0.3922, 0.5444, 0.6532, 0.7112, 0.7707, 0.8006, 0.8163 | 0.487 |
> > > |ODAM| 0.0000, 0.0000, 0.0217, 0.1718, 0.4815, 0.6835, 0.8021, 0.8662, 0.9055, 0.9248, 0.9380 | **0.580** |
> > > |Grad-CAM| 0.0000, 0.0000, 0.0008, 0.0106, 0.0349, 0.0704, 0.1105, 0.1531, 0.2088, 0.2557, 0.2928 | 0.114 |
> > > |Grad-CAM++| 0.0000, 0.0000, 0.0050, 0.0274, 0.0986, 0.1783, 0.2623, 0.3345, 0.4005, 0.4499, 0.4873 | 0.224 |
> > >
> > > |Pointing Game | PG in box $\uparrow$ | Energy based PG $\uparrow$| Compactness $\downarrow$ |
> > > |  ----  |  ----  | ----  | ---- |
> > > |BBAM| 0.9695 | 0.9693 | 0.0592 |
> > > |ODAM| **0.9878** | **0.9889** | **0.0432** |
> > > |Grad-CAM| 0.6306 | 0.6320 | 0.3316 |
> > > |Grad-CAM++| 0.6222 | 0.6334 | 0.2460 |
> > >
> > > Compared with BBAM, our ODAM performs better in terms of both faithfulness (Deletion/Insertion) and localization (PointGame/Compactness). Moreover, ODAM is over 350 times faster than BBAM; the average processing time (including model inference and heat map generation) of BBAM and ODAM for one prediction on GTX 1080Ti is 81.32s vs. 0.21s.
> > >
> > > Qualitatively, compared to ODAM, the BBAM heat map visualizations are sparser and more focused on a minimal set of keypoints, which is due to the L1 loss of the mask. For example, while ODAM highlights all the distinctive features of the head of a horse (mouth, nose, jawline, ears, eyes, forehead), BBAM will highlight only the ears and mouth.
> > >
> > > We will add these quantitative results and visual comparisons to the final paper in the Appendix.

---

> > > ### Author Response · Authors · 2022-12-08
> > > **User study for explaining bounding box regression using different gradient information**
> > >
> > > As discussed in Appendix A.5.6, there are 4 ways to visualize the explanations for the bounding box regression, which uses different gradient information. We have conducted a new user study to find out which version is most useful for users to understand how the detector has predicted the bounding box. In the questionnaire, for the predicted bounding box of an object, we show users the four kinds of explanation maps generated by ODAM as in the Appendix A.5.6:
> > > a)	positive gradients with ReLU showing the regions important for expanding the box;
> > > b)	negative gradients with ReLU showing the regions important for shrinking the box;
> > > c)	positive gradients without ReLU showing a summary of regions important in either direction, using different colors;
> > > d)	positive gradients with Abs (instead of ReLU) showing a summary of the regions important overall for the bounding box.
> > > Users are asked to rank these four kinds of maps from the best explanation (1st) to the worst explanation (4th) with respect to the predicted bounding box. The questionnaire is composed of 20 questions with five for each box attribute (left, top, right, bottom), and was taken by 10 users, resulting in a total of 200 responses. Here is the table of the percentage of ranking and average rank for each type of explanation map:
> > >
> > > | Map type | 1st | 2nd | 3rd |4th | Averaged Rank |
> > > |  ----  |  ----  | ----  | ---- | ---- |  ---- |
> > > | a) Pos w/ ReLU | **37.5%** | 32.0% | 17.5% | 13.0% | **2.06** |
> > > | b) Neg w/ ReLU | 11.5% | 17.0% | 30.5% | 41.0% |3.01|
> > > | c) Pos w/o ReLU | 32.0% | 19.0% | 27.5% | 21.5% | 2.39|
> > > | d) Pos w/ Abs | 19.0% | 32.0% | 24.5% | 24.5% | 2.55|
> > >
> > > The results indicate that the explanation map with only positive gradients better shows the user how the model predicted the bounding box, while the map with only negative gradients achieved the worst average rank (Wilcoxon signed-rank test, p<0.001). The average rank of Pos w/ ReLU is significantly better than the next best method, Pos w/o ReLU (Wilcoxon signed-rank test, p=.004). These results confirm that the positive gradients version of the bounding box explanation map is the most useful for users. We will add these results to Appendix A.5.6.

---

### Official Review · Reviewer_ZW78 · 2022-10-25

**Confidence:** 4
**Correctness:** 3
**Technical Novelty And Significance:** 3
**Empirical Novelty And Significance:** 3
**Recommendation:** 8

**Clarity, Quality, Novelty And Reproducibility:**

- Novelty

The paper proposes an approach for generating instance specific white box explanations of object detection that can be applied to one-stage and two-stage architectures. As far as I know, this is the first study about this kind of problem. However, the approach is a simple restriction to the detected instance of well known feature weighting explanation generation such as GradCAM: the main innovation compared to this family of explanation is to have introduced a specific training loss that is expected to improve explanation.


- Clarity and quality

The paper is clearly written and easy to follow. The experiments are rather convincing.

There are however a few points that need to be clarified.

** Miscellaneous questions and comments

Does the model learned with the ODAM-Train loss have the same performance as the original algorithm, what is the influence of adding the new loss ?

Did you compare the results using $H_{IoU}$ or $H_{class}$ only? Or without using the ground truth bounding box to compute the IoU term in $H_{comb}$ ? As a complementary comment, I found it quite strange to use the ground-truth to build the explanation for $H_{IoU}$ and not for $H_{class}$.

In the same vein, does ODAM-NMS exploit the explanation that has been computed using the bounding box ground truth for $H_{IoU}$?

I didn’t understand how are computed the gradients $\partial Y_c / \partial A_k$ used to build the GradCAM and GradCAM++ explanations. What is $Y_c$, is it the predicted class of a given instance?

When using attention-based architecture, [1]  proposes another  white-box method to generate instance-level explanations for object detector predictions? Can the approach be applied to transformer architectures such as DETR [2], and what would be the difference ?

** Several issues in the writing

No appendix provided, although it was claimed to contain useful information (pseudo-code for ODAM-NMS for instance).

In Tab. 5, does the indicated computing time correspond to Odam-NMS + heatmap computation, or only NMS once heatmap are already computed ? Did you measure the computing time of heatmap only ?


- Reproducibility

The appendix is not provided and does not contain the pseudo-code for Odam-NMS as announced in the main paper.

The learning parameters of Odam-Train are not provided (hyperparameters, weighting of losses)

- References

[1] Chefer, H., Gur, S., & Wolf, L. (2021). Generic attention-model explainability for interpreting bi-modal and encoder-decoder transformers. In Proceedings of the IEEE/CVF International Conference on Computer Vision (pp. 397-406)

[2] Carion, N., Massa, F., Synnaeve, G., Usunier, N., Kirillov, A., & Zagoruyko, S. (2020). End-to-end object detection with transformers. In European conference on computer vision (pp. 213-229). Springer, Cham.


**Strength And Weaknesses:**

== Strengths ==
- One of the few white-box  instance specific explanation generation for object detection
- The explanation combines class-specific and object localization information in a single heat-map
- ODAM-Train generally improves the object localization
- Exploits explanation to improve the performance of NMS (but slightly) for crowded scenes


== Weaknesses ==
- The explanation is not neutral since the model is modified by an extra loss term during learning.
- Mixing localization and class-specific information in a single representation may lead to confusion about the explanandum (what is explained).
- The computation of the explanation requires the object bounding box ground-truth.




**Summary Of The Paper:**

The authors propose a gradient-weighted visualized explanation technique for object detectors called ODAM. The technique produces instance-specific heat maps indicating the regions that have an impact on the object prediction. It combines attributions from the class and object localization prediction to produce a single heatmap that is used during the training.

To address the problem of crowded scenes and avoid potential heatmap “leaks” onto neighboring objects, the authors propose a method called ODAM-Train to better identify and discriminate the detected object from its neighbors. Two terms are added in the original learning loss function to constraint the consistency and separations of the detected objects.

To mitigate the NMS post-processing problems that still occur in crowded scenes with ODAM-Train, the authors propose ODAM-NMS. ODAM-NMS uses both the IoU between two predictions, and the heatmap correlations to evaluate the probability that they correspond to the same object.

The approach is evaluated on MS COCO dataset using various fidelity and accuracy metrics for explanation and shows good results compared to D-RISE.



**Summary Of The Review:**

The paper proposes a novel approach for generating instance specific white box explanations of object detection that can be applied to many current architectures. The evaluation on  MS-COCO gives good results on several metrics. There are however a few points that need to be clarified to recommend acceptance.

---

> ### Author Response · Authors · 2022-11-16
> **Response to Reviewer ZW78 -- part1:**
>
> Thanks for your insightful review. We have answered your concerns/questions here, and are happy to discuss it further. We have also updated the paper accordingly.
>
> **Q1:** *The explanation is not neutral since the model is modified by an extra loss term during learning.*
>
> **A1:** We think that our ODAM explanation method itself is neutral, since there is no training stage for ODAM, and it directly calculates the explanation from the model parameters. The “extra loss” term that the reviewer refers to is Odam-Train, where during training, we use ODAM to generate explanations and add a loss to encourage these explanations to be distinct for different objects. At inference time, the explanations are generated as usual with ODAM in a neutral manner.
>
> The question is whether specifically training a model to improve explainability is fair. We think that we are using a fair setting because the explanations are generated from the **whole** model that performs the object detection task, and thus supervising the explanations will also supervise the detector itself. Therefore, changes in the explanation maps from learning (e.g., better localized on the object) are reflected as changes in the detector’s strategy (using features on the object, and ignoring features elsewhere). In analogous work, researchers in VQA also train their models to produce better explanations; the bottom-up or top-down attention maps [A,B,C,D] of the VQA models are supervised with human eye gaze so that the VQA model focuses on those specific features to predict better sentences for visual reasoning. An example of a less fair setting is if the explanation heat map is predicted in a multi-task framework, where a separate NN head predicts the explanation map. In this case, training the explanation map head will mainly affect the common feature extractor, and has minimal effect on the detector head itself. Thus the disconnection between the generated heat map and the detector output is a major concern. Indeed improving explainability of models is a relatively new area, and we agree that care needs to be taken in the framework design. In summary, we believe that our design is fair because the supervision of the explanation map affects all layers of the detector via the top-down gradient computation of ODAM. We have added a sentence about this at the end of Sec 3.2.
>
> Please also see the separate response thread to all reviewers about object discrimination and Odam-Train.
>
> [A] D. H. Park, L. A. Hendricks, Z. Akata, A. Rohrbach, B. Schiele, T. Darrell, and M. Rohrbach, “Multimodal explanations: Justifying decisions and pointing to the evidence,” in CVPR, 2018, pp. 8779–8788.
>
> [B] T. Qiao, J. Dong, and D. Xu, “Exploring human-like attention supervision in visual question answering,” in AAAI, vol. 32, no. 1, 2018.
>
> [C] R. R. Selvaraju, S. Lee, Y. Shen, H. Jin, S. Ghosh, L. Heck, D. Batra, and D. Parikh, “Taking a hint: Leveraging explanations to make vision and language models more grounded,” in ICCV, 2019, pp. 2591–2600.
>
> [D] S. Chen, M. Jiang, J. Yang, and Q. Zhao, “Attention in reasoning: Dataset, analysis, and modeling,” IEEE Transactions on Pattern Analysis and Machine Intelligence, 2021.
>
> **Q2:** *Mixing localization and class-specific information in a single representation may lead to confusion about the explanandum (what is explained).*
>
> **A2:** We compute and visualize the combined heat map (Hcomb) in order to show an explanation from a holistic viewpoint, to allow easier comparison between the explanation maps generated with different models, in order to investigate the overall commonalities and differences. An example is shown in Fig. 3 right (“Comb”).  On the other hand, our framework *fully supports explanations of individual detector outputs* (class probability and bounding box coordinates), as shown in Fig. 3 left. Furthermore, all the quantitative evaluations and user studies are based on the explanation of individual detector output (class probability).
>
> **Q3:** *The computation of the explanation requires the object bounding box ground truth.*
>
> **A3:** Only the IoU heat map Hiou, which is used in the combined heat map Hcomb, uses the ground-truth bounding box. The other explanation maps, including for the bounding box coordinates (x1, x2, y1, y2) do not require the GT bounding box. Furthermore, **all** the quantitative evaluations in the experiments are based on heat maps of the object’s predicted class score (Hclass), and thus do not use the ground-truth bbox.
>
> We agree that using the GT box for Hcomb is confusing and unnecessary, so in the revised paper we have changed the definition of Hcomb to combine the heat maps of individual outputs: Hcomb = max(H_class, H_x1, H_x2, H_y1, H_y2), which does not use any GT. The relevant text appears in Sec 4.1, first paragraph. We have updated the visualizations in Fig 3, and the conclusions from this visualization are the same.

---

> ### Author Response · Authors · 2022-11-16
> **Response to Reviewer ZW78 -- part2:**
>
> **Q4:** *Did you compare the results using Hiou or Hclass only? Or without using the ground truth bounding box to compute the IoU term in Hcomb? As a complementary comment, I found it quite strange to use the ground-truth to build the explanation for Hiou and not for Hclass.*
>
> **A4:** In the quantitative evaluations, we only use Hclass, the heat map generated for the predicted class, since our purpose is to evaluate the quality of the identified important regions for identifying the object class. This also allows fair comparison with Grad-CAM and Grad-CAM++.
>
> To avoid confusion, in the revised manuscript we have removed the Hiou term, since it depends on the GT bounding box, and redefined Hcomb as stated in Answer A3 above.
>
> **Q5:** *In the same vein, does ODAM-NMS exploit the explanation that has been computed using the bounding box ground truth for Hiou?*
>
> **A5:** Odam-NMS uses the explanation map of the objects’ predicted class’s score (Hclass), since the goal is to find duplicate detections of the same object through their similar features used for classification. Odam-NMS also uses the IoU between *predicted* objects’ bboxes. Odam-NMS does **not** use the IoU of the GT bbox.
>
> **Comment about Q2-Q5:** In summary, the GT bounding box is not involved in any of the quantitative evaluations or user studies involving ODAM, Odam-Train, or Odam-NMS.  The Hiou was only used for a holisitic visualization of different detectors in Fig. 3 right. We apologize for the confusion, and in the updated paper, we have redefined Hcomb to not use the GT bounding box, and updated the corresponding Fig. 3 right. We have also made the targets for each heat map clear.
>
> **Q6:** *Does the model learned with the ODAM-Train loss have the same performance as the original algorithm, what is the influence of adding the new loss?*
>
> **A6:** The model trained with Odam-Train has similar performance to the baseline model in terms of detection accuracy. Using Odam-Train has negligible effect on the detector accuracy (AP and recall), while it boosts the object discrimination ability of the explanation maps. Please see Tables 8 and 9 in the Appendix A.4.3. On the other hand, using Odam-Train will make the faithfulness metrics worse than ODAM w/o Odam-Train, but still better than D-RISE (see Table 1) in Insertion and VEA. Interestingly, there seems to be a tradeoff between faithfulness and object discrimination, since the latter aims to only use pixels localized on the object. Future work could consider how to diminish this tradeoff.
>
> **Q7:** *I didn’t understand how are computed the gradients used to build the GradCAM and GradCAM++ explanations. What is Yc, is it the predicted class of a given instance?*
>
> **A7:** Yes, for GradCAM and GradCAM++, the gradients are calculated based on the output score of the predicted class Yc of the detected object instance.  Even though we use the class score of one particular object, the GradCAM explanation map will highlight all instances of the same class because of the channel-wise importance weighting when aggregating the activation maps, which ignores the spatial information, i.e., the location of the important feature. The details are added to Sec 4.2, 1st paragraph.

---

> ### Author Response · Authors · 2022-11-16
> **Response to Reviewer ZW78 -- part3:**
>
> **Q8:** *When using attention-based architecture, [1] proposes another white-box method to generate instance-level explanations for object detector predictions? Can the approach be applied to transformer architectures such as DETR [2], and what would be the difference?*
>
> *[1] Chefer, H., Gur, S., & Wolf, L. (2021). Generic attention-model explainability for interpreting bi-modal and encoder-decoder transformers. In Proceedings of the IEEE/CVF International Conference on Computer Vision (pp. 397-406)*
>
> *[2] Carion, N., Massa, F., Synnaeve, G., Usunier, N., Kirillov, A., & Zagoruyko, S. (2020). End-to-end object detection with transformers. In European conference on computer vision (pp. 213-229). Springer, Cham.*
>
> **A8:** Please see the separate response thread to all reviewers about applying ODAM to transformer-based detectors.
>
> **Q9:** *In Tab. 5, does the indicated computing time correspond to Odam-NMS + heatmap computation, or only NMS once heatmap is already computed? Did you measure the computing time of heatmap only?*
>
> **A9:** Table 5 reports the average processing time of one image, which includes the detector inference, heat map calculation, and NMS algorithm (NMS, Soft-, Feature-, or Odam-NMS). The heatmap computation takes about 2 ms per prediction (see Sec 3.1, last paragraph). In Odam-NMS, maps for multiple predictions are calculated in a batch. We have clarified the timing in the caption of Table 5.
>
> **Q10:** *The learning parameters of Odam-Train are not provided (hyperparameters, weighting of losses)*
>
> **A10:** We use equal weighting for the Odam-Train hyperparameters (see line after Eq. 3). Here we have added more details about the final loss used for training. Implementation details have been added to Appendix A.1.

---

> ### Comment · Reviewer_ZW78 · 2022-12-09
> **Final comments after rebuttal**
>
> Dear authors,
>
> Thank you for providing detailed answers to my concerns which have been addressed satisfactorily for most of them. I find the approach interesting and deserving publication and will upgrade my rating.
>
> Here are a few remaining comments.
>
> * I  appreciate the new experiments on DETR using ODAM and the comparison with the self-attention maps. The comparison is however qualitative on cherry picked samples. Visually, I find on the displayed examples that self-attention features already contain some kind of explanation that is quite comparable to the heat-map proposed by ODAM. This is also why I regret that, on the specific case of transformer-based models such as DETR, ODAM has not been compared to (Chefer et al.,2021) work as suggested.
>
> * Regarding the discussion about the fact that ODAM is not neutral: I was only wondering if the new detector w/ODAM would be affected by the new "explainability" loss, and table 8 in the appendix gave me the answer.
>
> * For "object discrimination", as you call it in your general answer, the fact that the existence of other objects in the scene has a contextual impact on the detection of others is for me not a "spurious correlation" but a desirable phenomenon, if it exists, to explain and reveal: natural vision makes use of such priming and primer effects and a good explanation of detection should take this into account if the algorithm under study exploits such a strategy. To assert "that good object discrimination will tend to use features only on the object" is very limited as a principle of explainability.

---

> > ### Author Response · Authors · 2022-12-10
> > **Appreciate your updated score**
> >
> > Thanks for your support!
> >
> > - Regarding DETR, we plan to do a thorough study of applying ODAM on transformer-based detectors as our future work. There are several parts of DETR that could be probed with ODAM, e.g., the position embedding, self-attention mechanism, etc. Thanks for the suggestion, and we will compare with (Chefer et al 2021) in our future work.
> >
> > - Yes, it is an interesting point. Both human vision and machine vision use context information, and our ODAM can reveal this since we do not artificially limit the heat map calculation to only the bounding box. We will add a few sentences discussing this in the paper.

---

### Author Response · Authors · 2022-11-05
**Upload the version with Appendix**

Dear AC and Reviewers,

Sorry, we had forgotten to upload the Appendix before the paper deadline (mistakenly assuming that it was one week after the main paper deadline). We have just now uploaded the Appendix.

We will address the reviewer's comments and update the paper/appendix later.

Thanks for your time,
the Authors

---

> ### Author Response · Authors · 2022-11-16
> **An updated version of paper**
>
> Dear reviewers and AC:
>
> We have uploaded a revised paper and appendix to address the reviewers' insightful comments. The modifications are marked in blue text in the paper and appendix. The major changes are:
>
> 1. Add visualization with DETR in Figure 3 right. More visualizations are put in Appendix A.5.7. (Reviewer ZW78 Q8; Reviewer EVbq Q5; Reviewer y3xs Q5)
> 2. We add a user study on object discrimination to show that Odam-Train indeed improves users’ confidence in object discrimination from the heat map. The summary is added to Table 3(b) and Sec 4.2 Object Discrimination, and the details are added to Appendix A.3 and Table 6. These results also show that ODI is correlated with the user’s object discrimination, and thus ODI is a valid metric for measuring object discrimination. (Reviewer EVbq Q2; Reviewer y3xs Q1b, Q2b, Q2c, Q4c)
> 3. Added a sentence about the fairness of supervising the explanation maps at the end of Sec 3.2. (Reviewer ZW78 Q1)
> 4. Add the faithfulness results with Odam-Train to Fig. 5 and Tab.1, and discussion in Sec 4.2. (Reviewer EVbq Q2b).
> 5. Added discussion about object discrimination and context (Sec 5). (Reviewer y3xs Q2a)
> 6. To avoid using Hiou which uses the GT bounding box, we have redefined Hcomb to use the heat maps for (Yc, x1, x2, y1, y2), which do not use any GT bounding box. The description of Hcomb is moved to the experiments, Sec 4.1 1st paragraph, and Fig. 3 is updated. (Reviewer ZW78 Q2, Q3; Reviewer K7eB Q1a)
> 7. Clarified the meaning of (x1,y1,x2, y2) and its gradient relationship in Fig. 3 caption, and added additional visualizations of bounding box regression in Appendix A.5.6 and Fig. 16. (Reviewer EVbq Q3).
>
> We have also added more visualizations to the Appendix (some of which were mistakenly not uploaded before):
>
> 8. [in previous appendix] Show comparison of Odam-NMS w/ and w/o Odam-Train in Appendix 4.3 (Reviewer y3xs Q3b; Reviewer ZW78 Q6)
> 9. [in previous appendix] Show additional visualizations in Appendix A.5 (Reviewer y3xs Q4a)
> 10. Add comparison of heat maps generated with different feature map layers in A.5.3. (Reviewer K7eB Q4)
> 11. [in previous Appendix] Explore the failure modes of the detector in Appendix A.5.2 (Figs 11 and 12). (Reviewer y3xs Q1a, Q4b)
>
> Finally, we have improved the presentation as suggested by the reviewers:
>
> 12. In various places, make the target of the explanation heat maps to be clearer. (Reviewer ZW78 Q4, Q5, Q7; Reviewer K7eB Q3)
> 13. Added missing references to Section 2. (Reviewer EVbq Q4, Reviewer y3xs Q3a)
> 14. Update the description of the object discrimination explanation task to be clearer (Sec 1). (Reviewer y3xs Q1b, Q2a)
> 15. Other improvements in presentation (Reviewer ZW78 Q9, Q10; Reviewer y3xs Q1b, Q6, Q7)
> 16. Light editing to keep the main paper within the page limit.
> 17. We fixed a bug in computing the deletion scores in Table 1 and Figure 5(a). The conclusions from the figure are the same as before.
>
> We hope that the revised version of the paper and appendix well address the reviewers' comments. For responses to specific comments, please see the response threads. If there are any remaining concerns, please let us know.
>
> Thanks for your time,
>
> -the Authors

---

### Author Response · Authors · 2022-11-16
**Common concern about object discrimination and Odam-Train**

**Reviewer y3xs Q2a:** The motivation for Odam-Train and the Object Discrimination Index seem flawed. As mentioned above, the paper seems to hint at compactness of heatmaps being an indicator for a good visual explainability tool. But I do not quite understand why these two things are correlated. What if the existence of one instance of category A makes it more likely that another instance of the same category exists in the image? Another example: If the image captures the sun, wouldn't that influence the probability (confidence scores) of predicting the category "umbrella"?

**Reviewer EVbq Q2:** I am not sure if ODAM-Train can provide "real" explanation of the neural network. It is natural for different predictions to produce different interpretations, but ODAM-Train forces the explanations of different predictions to be consistent. Although this may improve several numerical results, it is questionable whether it can provide real explanations of neural networks.

**Reviewer y3xs Q2b:** Odam-Train essentially has the goal to make the heatmaps more compact. The loss functions make sense for that goal. But I do not understand how this makes ODAM a better explainability tool? Obviously, if we consider compactness of heatmaps as good for explaining a model, then optimizing for that will make it better. But does this really improve explainability of the model?

**Reviewer y3xs Q2c:** The same argument is true for the proposed Object Discrimination Index, as a quantitative measure for how good the visual explanation is. I'm wondering whether the motivation for this metric is valid.

**Reviewer ZW78 Q1:** The explanation is not neutral since the model is modified by an extra loss term during learning.

---

> ### Author Response · Authors · 2022-11-16
> **Answer--part1:**
>
> Here we will give a summary answer to these concerns. For specific and detailed answers, please see the responses to individual reviewers.
>
> For object detectors, we consider two types of explanation tasks: 1) the traditional explanation task of “what context/features are important for the class prediction?”, which we can denote as “*object specification*”; 2) the detector-specific explanation of “which object was actually detected?”, which we denote as “*object discrimination*”. For object discrimination, intuitively we would like the heat map to be well-localized on the target object and have negligible values on the other objects. In this way, the user is certain that the correct object was localized when looking at the heat map. We hinted this as “compactness” in the paper, but the term “well-localized” is better, and we have changed it in the revised paper.
>
> A detector with good object discrimination will tend to use features only on the object, and thus not use features on other objects (i.e., context) as much as other detectors. This is not necessarily a problem since the detector that does not use context information could generalize better to various scenarios, and is less susceptible to spurious correlations in the training dataset (e.g., objects always co-occurring during training).
>
> In the experiments, we find that our ODAM generates explanations with better object discrimination (measured via ODI) than other methods. We also conduct a new user study on object discrimination (see Table 3a and Sec 4.2 Object Discrimination, and Table 6 in Appendix A.3), which shows that ODAM also induces higher accuracy and higher confidence in users when they identify the detected object from the heat map, i.e., they have higher confidence in the detector’s explanation about which object was detected.
>
> To improve the object discrimination of the detector, we propose Odam-Train, which encourages the heat maps for detections of the same GT object to be consistent, and those of different objects to be distinct. In this way, the detector learns to focus more on the object itself to improve object discrimination. Table 3a shows that using Odam-Train improves the object discrimination (measured via ODI), compared to without it. The new user study (Table 3b and Table 6) also shows that Odam-Train improves user confidence to discriminate the object in the heat map from other objects. Therefore, we think that Odam-Train is indeed providing real explanations that improve user confidence in the detector.
>
> Using Odam-Train has a negligible effect on the detector accuracy, as shown in Tables 8 and 9 in Appendix A.4.3. On the other hand, using Odam-Train will make the faithfulness metrics slightly worse than ODAM w/o Odam-Train, but still better than D-RISE (see Table 1). Interestingly, there seems to be a tradeoff between faithfulness and object discrimination, since the latter aims to only use pixels localized on the object. Future work could consider how to diminish this tradeoff.

---

> ### Author Response · Authors · 2022-11-16
> **Answer--part2:**
>
> Finally, there is a question as to whether specifically training a model to improve explainability is fair. We think that we are using a fair setting because the explanations are generated from the **whole** model that performs the object detection task, and thus supervising the explanations will also supervise the detector itself. Therefore, changes in the explanation maps from learning (e.g., better localized on the object) are reflected as changes in the detector’s strategy (using features on the object, and ignoring features elsewhere). In analogous work, researchers in VQA also train their models to produce better explanations; the bottom-up or top-down attention maps [A,B,C,D] of the VQA models are supervised with human eye gaze so that the VQA model focuses on those specific features to predict better sentences for visual reasoning. An example of a less fair setting is if the explanation heat map is predicted in a multi-task framework, where a separate NN head predicts the explanation map. In this case, training the explanation map head will mainly affect the common feature extractor, and has minimal effect on the detector head itself. Thus the disconnection between the generated heat map and the detector output is a major concern. Indeed improving explainability of models is a relatively new area, and we agree that care needs to be taken in the framework design. In summary, we believe that our design is fair because the supervision of the explanation map affects all layers of the detector via the top-down gradient computation of ODAM.
>
> [A] D. H. Park, L. A. Hendricks, Z. Akata, A. Rohrbach, B. Schiele, T. Darrell, and M. Rohrbach, “Multimodal explanations: Justifying decisions and pointing to the evidence,” in CVPR, 2018, pp. 8779–8788.
>
> [B] T. Qiao, J. Dong, and D. Xu, “Exploring human-like attention supervision in visual question answering,” in AAAI, vol. 32, no. 1, 2018.
>
> [C] R. R. Selvaraju, S. Lee, Y. Shen, H. Jin, S. Ghosh, L. Heck, D. Batra, and D. Parikh, “Taking a hint: Leveraging explanations to make vision and language models more grounded,” in ICCV, 2019, pp. 2591–2600.
>
> [D] S. Chen, M. Jiang, J. Yang, and Q. Zhao, “Attention in reasoning: Dataset, analysis, and modeling,” IEEE Transactions on Pattern Analysis and Machine Intelligence, 2021.

---

### Author Response · Authors · 2022-11-16
**Common questions about applying ODAM to transformer-based detectors**

**Reviewer ZW78 Q8:** When using attention-based architecture, [1] proposes another white-box method to generate instance-level explanations for object detector predictions? Can the approach be applied to transformer architectures such as DETR [2], and what would be the difference?

[1] Chefer, H., Gur, S., & Wolf, L. (2021). Generic attention-model explainability for interpreting bi-modal and encoder-decoder transformers. In Proceedings of the IEEE/CVF International Conference on Computer Vision (pp. 397-406)

[2] Carion, N., Massa, F., Synnaeve, G., Usunier, N., Kirillov, A., & Zagoruyko, S. (2020). End-to-end object detection with transformers. In European conference on computer vision (pp. 213-229). Springer, Cham.

**Reviewer EVbq Q5:** Following recent trends, I recommend the authors apply the method to transformer-based object detectors like DETR. In addition, the transformer-based methods provide attention map, which also can be considered as an interpretation map. Please provide the detailed comparison with those methods.

**Reviewer y3xs Q5:** The paper mentions that ODAM is applicable to one-stage detectors like YOLO or FCOS. I am missing a discussion about recent Transformer-based (DETR-based [C]) methods like DINO [A]?

[A] “DINO: DETR with Improved DeNoising Anchor Boxes for End-to-End Object Detection.” Zhang et al. arXiv 2203.03605; [C] “End-to-end object detection with Transformers.” Carion et al. ECCV 2020

**A:** Yes, our approach can be applied to transformer-based detectors. There are two ways to use the transformer in object detection: 1) using the vision transformer-based architecture (e.g., ViT) as a backbone for extracting features, followed by a standard CNN detector pipeline (neck and head); 2) using the transformer as the detector head to build a set prediction detector (DETR), where the backbone is a standard CNN feature extractor (e.g. ResNet50).

For the first one (ViT backbone), we have already visualized the detector with pyramid vision transformer (PVT) as backbone in Fig. 3 (right).

For the second one (DETR), we perform ODAM on DETR with the features from the ResNet50 backbone, and also add the result to Fig. 3 right (more examples in Fig 17). Interestingly, according to the ODAM heat maps, DETR more heavily focuses on object parts, while ignoring other regions, compared to other detectors. This perhaps is a consequence of using the patch-set representation and transformer for the DETR head.

Regarding the attention map of the transformer in DETR and its variants, the transformer self-attention mechanism produces a bottom-up attention map (*i.e., generally which features are interesting*). In contrast, the ODAM generates a top-down attention map (*i.e, which features are important for the output prediction*). It should be noted that for bottom-up attention, even if a feature is highlighted in the attention map, there is no guarantee that the feature is actually used in the subsequent output prediction. In contrast, for the top-down attention, all highlighted features should be important for generating the prediction. Therefore, we think that top-down visual explanations are still necessary for transformer-based detectors [D]. In the Appendix A.5.7 and Fig. 17 we have visualized more ODAM visual explanations for DETR compared with other detectors. We are currently working on visualizing the self-attention in DETR and will add it later.

In future work, we will do a further detailed study on further applying ODAM to transformer-based detectors, especially under the interpretation as an object-part relationship learner. Although the transformer-based detectors are promising, CNN-based 1-stage and 2-stage detectors are still widely used in both academia and industry, and thus visual explanation methods for them are still needed.

[D] Jain, S., & Wallace, B. C. (2019). “Attention is not explanation.” In Proceedings of the 2019 Conference of the North American Chapter of the Association for Computational Linguistics: Human Language Technologies, pp. 3543–3556.

---

> ### Author Response · Authors · 2022-11-18
> **Adding comparison with attention maps in transformer**
>
> Dear reviewers and AC:
>
> We have just added a new Appendix A5.8 and Figures 18 and 19, which compare the explanation maps for DETR using ODAM and the visualizations of self-attention weights in both the encoder and decoder of DETR. From the visualizations, we obtain some interesting interpretations of DETR: 1) the encoder self-attention mainly highlights the information related to object and context, but not all the context regions highlighted by self-attention are actually used for prediction according to the ODAM class heat map; 2) the decoder self-attention mainly highlights all regions at the extremities of the object (i.e., near the bounding box), but cannot distinguish which regions are important for a particular box coordinate. In contrast, using ODAM explanations can visualize which features are particularly important for each predicted attribute (class and each bounding box coordinate). How to utilize the top-down ODAM heat maps and the bottom-up self-attention maps for generating visual explanations is an interesting area of future work.
>
> Thanks for your time,
>
> -the Authors

---

### Author Response · Authors · 2022-11-27
**Looking forward to your responses or further suggestions/comments!**

Dear reviewers and AC:

Thanks again for all of your constructive suggestions, which have helped us improve the quality and clarity of the paper!
Since we posted our rebuttal for a while, we have not heard any post-rebuttal response yet.
Please don’t hesitate to let us know if there are any additional clarifications or experiments that we can offer, as we would love to convince you of the merits of the paper. We appreciate your suggestions. Thanks!

-the authors

---

### Decision · Program_Chairs · 2023-01-20

**Decision:**

Accept: poster

**Justification For Why Not Higher Score:**

As described in detail below, the main value of this paper lies in the combination of a few simple (but solid) ideas and their evaluation, while the fundamental novelty is limitted. Therefore, the AC supposes that it is a bit short for spotlight or oral.

**Justification For Why Not Lower Score:**

This paper initially received the score of 5,5,6,6. Since the supplementary material was forgot by the authors to be uploaded, many details were unclear. The reviewer raised many concerns regarding its novelty, insufficient experiments and analysis, and unclear details of the method.
During the discussion period, the authors and the reviewers had very active discussion. The authors have presented many detailed explanations of the method/experiments and additional results as well as the improved paper and supplementary. Notably, novel experiments showing the effectiveness of ODAM on Transformer-based models have been presented, which was one of the main concerns in the beginning. Although not 100% satisfied, all reviewers consider that their concerns are generally addressed and raised their scores to 6,6,6,8.
Meanwhile, the concern regarding the fundamental novelty may still remain. Although the AC agrees that the novelty of plain ODAM is not very big, the paper also provides two extension modules, OdamTrain and Odam-NMS, which are reasonable and not trivial. Also, they provide insight into how to use the visual explanation to boost detection performance, which is interesting and will stimulate not only the XAI community but also the broader audience in visual recognition. Overall, the reviewers are unanimous for acceptance, and the AC also conclude that the pros of the paper sufficiently overweight the cons, thus recommends acceptance.

**Metareview: Summary, Strengths And Weaknesses:**

Summary:
This papers a method of instance-specific visual explanation for object detection called ODAM. ODAM follows the idea of gradient-based visualization from Grad-CAM, while the key differences are that the "importance-weights" are computed for each individual pixel and feature map to generate instance-based heatmaps. Because the heatmap may “leaks” onto neighboring objects in crowded scenes, the authors further propose a method called ODAM-Train that adds loss functions to constraint the consistency and separations of the detected objects. They also proposed ODAM-NMS which uses the correlations between instance-level heat maps and the box IoUs to remove duplicate proposals of the same object. In the experiments, the proposed approach is evaluated on COCO and CrowdHuman datasets using various fidelity and accuracy metrics for explanation and shows good results compared to D-RISE.

Strengths:
1. This paper investigates an important yet relatively unexplored topic, visual explanation for object detection.
2. The proposed method is simple but intuitive and works.
3. The experiments are thorough and show the effectiveness of the proposed method, using many detection models such as one- and twostage, and transformer-based detectors with different types of backbones and detector heads.
4. OdamTrain and Odam-NMS provide insight into how to use the visual explanation to boost detection performance.

Weaknesses:
1. Technical novelty is not very big since the core idea is a relatively straightforward extension of Grad-CAM.
2. Experiments on Transformer-based models (also, comparison with self-attention-based explanation methods) can be elaborated more.


**Note From Pc:**

if the above contains the word "oral" or "spotlight" please see: "oral" presentation means -> notable-top-5% and "spotlight" means -> notable-top-25%. As stated in our emails, we are disassociating presentation type from AC recommendations